# TAG: Tangential Amplifying Guidance for Hallucination-Resistant Diffusion Sampling

## Abstract

Recent diffusion models achieve the state-of-the-art performance in image generation, but often suffer from semantic inconsistencies or *hallucinations*. While various inference-time guidance methods can enhance generation, they often operate *indirectly* by relying on external signals or architectural modifications, which introduces additional computational overhead. In this paper, we propose **T**angential **A**mplifying **G**uidance (**TAG**), a more efficient and *direct* guidance method that operates solely on trajectory signals without modifying the underlying diffusion model. TAG leverages an intermediate sample as a projection basis and amplifies the tangential components of the estimated scores with respect to this basis to correct the sampling trajectory. We formalize this guidance process by leveraging a first-order Taylor expansion, which demonstrates that amplifying the tangential component steers the state toward higher-probability regions, thereby reducing inconsistencies and enhancing sample quality. TAG is a plug-and-play, architecture-agnostic module that improves diffusion sampling fidelity with minimal computational addition, offering a new perspective on diffusion guidance.

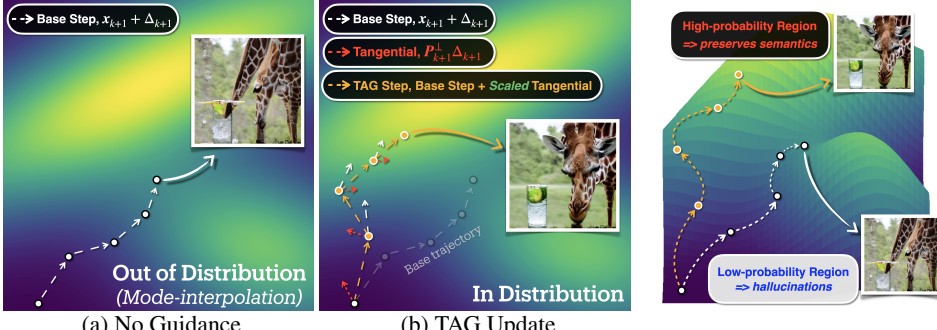

(a) No Guidance  (b) TAG Update

Figure 1: **Conceptual visualization of Tangential Amplifying Guidance (TAG)** from a mode-interpolation perspective (Aithal et al., 2024). Unlike **(a)** no guidance case, **(b)** TAG decomposes the base increment $\Delta_{k+1}$ on the latent sphere into parallel $P_{\mathcal{M}_{k+1}}\Delta_{k+1}$ and orthogonal (i.e., tangential) $P^{\perp}_{\mathcal{M}_{k+1}}\Delta_{k+1}$ components (equation 7). By preserving the parallel component while adding a *scaled* tangential component, TAG isolates the data-relevant part of the update (§3) and can more effectively navigate the data manifold, leading to samples that contain more semantic structure. We make this precise by proving that amplifying the tangential has the effect of guiding the trajectories toward regions of higher model density while mitigating off-manifold drift (§4, equation 15).

## 1 Introduction

Hallucination in diffusion models refers to the phenomenon of generating samples that violate the data distribution or contradict conditioning, thus failing to provide meaningful outputs. For example, it often manifests as mixed-up objects (Okawa et al., 2023) or anatomically implausible structures (e.g., extra-fingers hands). Recent evidence suggests that the primary source of such errors lies in a failure mode known as mode interpolation. During sampling, trajectories may traverse low-density valleys between distinct modes of the data distribution, causing attribute mismatches and structural inconsistencies (Aithal et al., 2024).

A widely adopted remedy involves inference-time guidance strategies, such as classifier-free guidance (CFG) (Ho & Salimans, 2021) and their variants (Hong et al., 2023; Ahn et al., 2024a; Karras

et al., 2024a; Rajabi et al., 2025; Kwon et al., 2025; Sadat et al., 2025; Dinh et al., 2025; Hong, 2024). Under the assumption that deviating from low-probability regions enhances sample quality, most of these methods employ *residual scaling*, using the difference between the conditional and unconditional branches to guide the generation process away from the unconditional model's outputs. While effective, these mechanisms are fundamentally *indirect*: instead of navigating along the *intrinsic geometry* of the data distribution, they proceed by repeatedly moving away from an unconditional estimate at each step of the process.

In contrast, we propose a more efficient *direct* solution grounded in Tweedie's identity (Tweedie et al., 1984), which relates the score to the posterior mean under Gaussian corruption. This link motivates a decomposition of the model update based on its *intrinsic geometry*: a drift component that advances the radius along the prescribed noise schedule (i.e., noise level), and a tangential component that moves along the *data-manifold*, approximately preserving the overall *radius* while refining the sample's structure and semantics. We observe that the tangential component carries rich structural information (Figure 2), and amplifying it reduces out-of-distribution samples (Figure 3).

Drawing upon the principle of *amplifying the tangential component* during inference, we derive **T**angential **A**mplifying **G**uidance (**TAG**), a plug-and-play method that emphasizes the tangential component of the *score update*. TAG steers the sampling trajectory to follow the underlying data manifold closely. TAG integrates seamlessly with standard diffusion backbones—whether conditioned or not—without requiring additional denoising evaluations or retraining.

We can summarize our contributions as follows:

- We establish a concrete link between the score's intrinsic geometry and sample quality, proving that amplifying the tangential components of the scores steers sampling trajectories toward the in-distribution manifold.

- We introduce TAG, a computationally efficient and architecture-agnostic algorithm that realizes this geometric principle in practice.

## 2 PRELIMINARIES

**Score-based Diffusion Model.** Score-based generative models learn a time-indexed score function that approximates the gradient of the log-density of noise-perturbed data,

$$\boldsymbol{s}_\theta(\boldsymbol{x}, t_k) \approx \nabla_{\boldsymbol{x}} \log p(\boldsymbol{x} \mid t_k), \quad t_k \in \{t_K > \cdots > t_0\} \text{ denotes the } k\text{-th discretized timestep,}$$

to reverse a gradual noising process for sample generation. This approach provides a continuous-time framework that unifies earlier discrete-time Denoising Diffusion Probabilistic Models (DDPMs) (Sohl-Dickstein et al., 2015; Ho et al., 2020) through the lens of stochastic differential equations (SDEs) (Song et al., 2020b). The core idea involves a forward-time SDE that transforms complex data into a simple prior distribution, given by

$$d\boldsymbol{x} = \mathbf{f}(\boldsymbol{x}, t)dt + g(t)d\mathbf{W}.$$

Generation is then performed by the corresponding reverse-time SDE, which becomes tractable by substituting the unknown true score with the learned model $\boldsymbol{s}_\theta$ (Anderson, 1982). To solve this numerically, we discretize the time horizon over timesteps $t_k \in \{t_K > \cdots > t_0\}$. This score network, typically a noise-conditional U-Net, is trained efficiently via denoising score matching across various noise levels (Vincent, 2011; Song & Ermon, 2019). For sampling, one can use numerical methods like predictor-corrector schemes to simulate the stochastic reverse SDE, or solve an associated deterministic ordinary differential equation (ODE) known as the probability-flow ODE. This continuous-time framework not only provides a theoretical basis for widely used deterministic samplers like DDIM (Song et al., 2020a) but has also inspired modern refinements, such as the preconditioning and parameterization in EDM (Karras et al., 2022), which further enhance the trade-off between sample quality and efficiency.

**Inference-Time Guidance.** Numerous methods modify the update field during sampling to improve fidelity without retraining. Early CFG-style guidance (Ho & Salimans, 2021) steers samples by rescaling residual signals, and complementary approaches replace external cues with model-internal signals for guidance (Hong et al., 2023; Ahn et al., 2024a; Hong, 2024). However, prior analyses show that naïve, geometry-agnostic scaling can reduce diversity or perturb solver dynamics

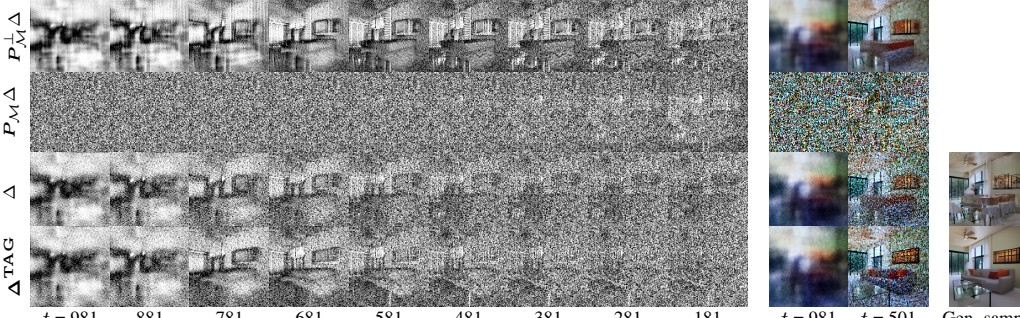

$t = 981 \quad 881 \quad 781 \quad 681 \quad 581 \quad 481 \quad 381 \quad 281 \quad 181 \qquad t=981 \quad t=501 \quad$ Gen. sample

Figure 2: **Amplifying the tangential component enhances semantic content by isolating it from noise.** This figure illustrates the decomposition of the update step $\Delta$ into *normal* and *tangential* components. Subtracting the unstructured, noisy normal component $\boldsymbol{P}_{\mathcal{M}}\Delta$ from the original update acts as a *denoising operation*, revealing the tangential component $\boldsymbol{P}_{\mathcal{M}}^{\perp}\Delta$, which preserves the *principal semantic structure*. Images decoded from intermediate timesteps ($t=981, 501$) indicate that semantic information is most salient in the tangential component. Motivated by this observation, our method $\boldsymbol{\Delta}^{\mathbf{TAG}}$ amplifies this semantically rich component, yielding a clearer and more coherent final sample (far right) than that obtained from the unmodified $\Delta$.

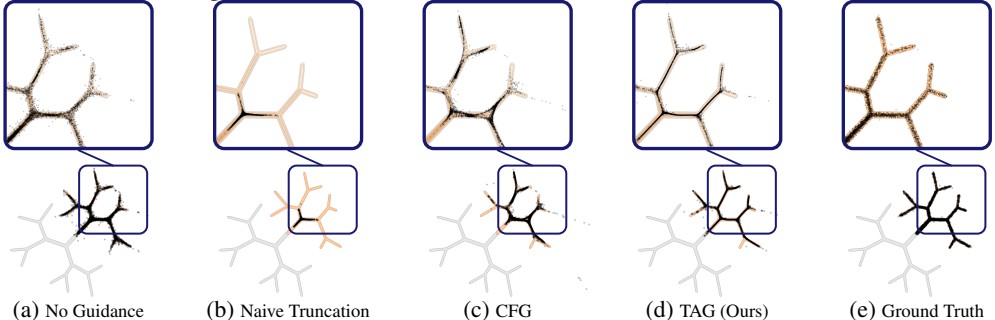

(a) No Guidance      (b) Naive Truncation      (c) CFG      (d) TAG (Ours)      (e) Ground Truth

Figure 3: **Sampling on a 2D branching distribution (Karras et al., 2024a) under different guidance methods.** (a) *No guidance*: probability mass drifts off the data manifold, yielding fragmented branches and OOD (Out of Distribution) points. (b) *Naive truncation*: suppresses some OOD but oversimplifies the geometry, dropping fine branches. (c) *CFG*: reduces boundary violations but also reduces diversity and can still leave OOD strays in our run. (d) **TAG (Ours)**: trajectories are steered toward high-density regions along the branches, suppressing off-manifold outliers while retaining detail. (e) *Ground truth*. Overall, TAG achieves the highest similarity to the GT distribution **without additional #NFEs**, concentrating mass on the correct branches while reducing OOD outliers.

(Dhariwal & Nichol, 2021; Kynkäänniemi et al., 2024). These limitations motivate geometry-aware guidance that asks not only how much to scale, but which directions to emphasize. Representative methods along this line use projections to dampen undesired components (e.g., high-scale saturation or cond–uncond mismatch) (Sadat et al., 2025; Kwon et al., 2025; Armandpour et al., 2023), or are developed for more specific problem settings such as loss-guided inverse problems or unpaired I2I (He et al., 2023; Sun et al., 2023) (see §B for more detailed discussion). Overall, these strategies integrate cleanly with modern solvers and effectively suppress off-manifold drift, but are often closely tied to particular guidance algebra or task-specific assumptions.

## 3 MOTIVATION AND INTUITION

Under Gaussian corruption, Tweedie's formula (Tweedie et al., 1984) links the posterior mean of the clean signal to the noisy observation via the score (i.e., the gradient of the log marginal density):

$$\mathbb{E}[\boldsymbol{x}_0|\boldsymbol{x}_k] = \Big( \underbrace{\boldsymbol{x}_k}_{:=\text{ drift term}} + \underbrace{\sigma_k^2 \nabla_{\boldsymbol{x}} \log p(\boldsymbol{x} \mid t_k)\big|_{\boldsymbol{x}=\boldsymbol{x}_k}}_{:=\text{ Tweedie increment } \Delta_k^{\mathrm{Tw}}, \ (a.k.a. \text{ data term})} \Big) / \sqrt{\bar{\alpha}_k}. \tag{1}$$

Geometrically, the score field $\nabla_{\boldsymbol{x}} \log p(\boldsymbol{x}|t_k)\big|_{\boldsymbol{x}=\boldsymbol{x}_k}$ points in the direction of steepest increase of the marginal density. Tweedie's formula therefore adjusts $\boldsymbol{x}_k$ in this ascent direction, nudging the state toward higher-probability regions. Therefore, the aim of modeling is to bias this movement toward *data-driven directions*.

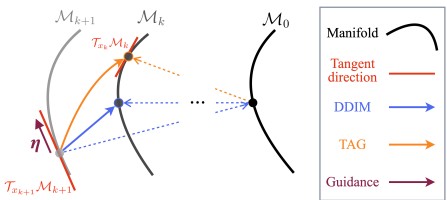

Figure 4: **Illustration of Tangential Amplification.** The curves $\mathcal{M}_k$ denote the noisy data manifolds at successive diffusion steps. TAG decomposes $\Delta_{k+1}$ into normal and tangential components with respect to $\mathcal{M}_{k+1}$, and amplifies the tangential component within the local tangent space $\mathcal{T}_{x_{k+1}}\mathcal{M}_{k+1}$ by a factor $\eta$.

To get better intuition for these *data-driven directions*, we appeal to the Gaussian annulus theorem (Blum et al., 2020), which states that an isotropic Gaussian in $\mathbb{R}^d$ concentrates most of its mass near a thin spherical shell. Since each corruption step adds such noise to the data, the corrupted distribution $p(\boldsymbol{x}|t_k)$ likewise places most of its mass near a thin shell. Consequently, we can regard the high-probability region of $p(\boldsymbol{x}|t_k)$ as a *noisy* data manifold $\mathcal{M}_k$ embedded near this shell, and thus our goal is to move along this high-probability region while keeping the radial component prescribed by the drift term unchanged. Therefore, to isolate the drift term, we define:

$$\boldsymbol{P}_{\mathcal{M}_k} = \widehat{\boldsymbol{x}}_k\widehat{\boldsymbol{x}}_k^\top, \quad \boldsymbol{P}_{\mathcal{M}_k}^\perp = I - \boldsymbol{P}_{\mathcal{M}_k} \quad \text{where} \quad \widehat{\boldsymbol{x}}_k := \boldsymbol{x}_k/\|\boldsymbol{x}_k\|_2, \tag{2}$$

each denotes the manifold-wise normal projection and tangential projection, respectively (Figure 4). Guided by this separation, we form the amplified state $\boldsymbol{x}^+$ via

$$\boldsymbol{x}^+ = \left(\boldsymbol{x}_k + \boldsymbol{P}_{\mathcal{M}_k}\Delta_k^{\text{Tw}}\right) + \eta\,\boldsymbol{P}_{\mathcal{M}_k}^\perp\Delta_k^{\text{Tw}}, \quad \text{with} \quad \eta \geq 1. \tag{3}$$

By doing so, we can preserve the radial first-order term (equation 19) while biasing the step toward *higher-probability regions*, $\nabla_{\boldsymbol{x}}\log p(\boldsymbol{x}|t_k)\big|_{\boldsymbol{x}=\boldsymbol{x}_k}$ (Empirical evidence is provided in Figure 2 and 3). In the following section (§4.1), we formalize this bias as a *constrained MLE* update that allocates first-order gain to the tangential subspace.

# 4    TAG: Tangential Amplifying Guidance

We introduce Tangential Amplifying Guidance (TAG), which reweights base increments along normal/tangential directions on the latent space.

**Definitions & Algorithm.** We work per sample on $\mathbb{R}^{C \times H \times W} \cong \mathbb{R}^d$ with Euclidean inner product $\langle \cdot, \cdot \rangle$ and norm $\|\cdot\|_2$. Let $\{t_k\}_{k=K}^0$ be *descending* timesteps with $t_K > \cdots > t_0$, and let $\epsilon_\theta$ denote the denoiser. Given $\boldsymbol{x}_{k+1}$ at time $t_{k+1}$, the denoiser predicts

$$\boldsymbol{\varepsilon}_{k+1} = \epsilon_\theta(\boldsymbol{x}_{k+1}, t_{k+1}).$$

A base solver (e.g., DDIM) then produces a *provisional* state (Karras et al., 2022)

$$\tilde{\boldsymbol{x}}_k = a_{k+1}\boldsymbol{x}_{k+1} + b_{k+1}\boldsymbol{\varepsilon}_{k+1}, \quad \text{where} \quad a_{k+1}, b_{k+1} \quad \text{are base solver coefficients.} \tag{4}$$

Corresponding base increment at $\boldsymbol{x}_{k+1}$ is defined as

$$\Delta_{k+1} := \tilde{\boldsymbol{x}}_k - \boldsymbol{x}_{k+1}. \tag{5}$$

For any $\boldsymbol{x} \in \mathbb{R}^d$, we define the unit vector and orthogonal projectors

$$\widehat{\boldsymbol{x}} = \boldsymbol{x}\,/\,\|\boldsymbol{x}\|_2, \qquad \boldsymbol{P}_{\mathcal{M}}(\boldsymbol{x}) = \widehat{\boldsymbol{x}}\widehat{\boldsymbol{x}}^\top, \qquad \boldsymbol{P}_{\mathcal{M}}^\perp(\boldsymbol{x}) = \boldsymbol{I} - \boldsymbol{P}_{\mathcal{M}}(\boldsymbol{x}). \tag{6}$$

Figure 4 illustrates this tangential–normal decomposition along the sampling trajectory. Given positive scales $\eta \geq 1$, TAG *reweights* the base increment at $\boldsymbol{x}_{k+1}$:

$$\boldsymbol{x}_k \leftarrow \boldsymbol{x}_{k+1} + \boldsymbol{P}_{\mathcal{M}_{k+1}}\Delta_{k+1} + \eta\,\boldsymbol{P}_{\mathcal{M}_{k+1}}^\perp\Delta_{k+1}$$
$$\text{where} \quad \boldsymbol{P}_{\mathcal{M}_{k+1}} = \boldsymbol{P}_{\mathcal{M}}(\boldsymbol{x}_{k+1}), \; \boldsymbol{P}_{\mathcal{M}_{k+1}}^\perp = \boldsymbol{P}_{\mathcal{M}}^\perp(\boldsymbol{x}_{k+1}). \tag{7}$$

## 4.1    Why does TAG improve Image Quality?

**Log-likelihood maximization.** A foundational goal of training generative models is to maximize the log-likelihood of the data, as formalized by the Maximum Likelihood Estimation (MLE) principle:

$$\max_\theta \sum_i \log p_\theta(\boldsymbol{x}_i).$$

This principle suggests that high-quality samples should concentrate in regions of high probability. To connect this idea to an *update rule*, we relate likelihood increase to movement along the score

---

**Algorithm 1** Tangential Amplifying Guidance (TAG)

---

**Require:** Denoiser $\epsilon_\theta(\cdot)$, timesteps $\{t_k\}_{k=K}^0$, base solver coefficients $a_{k+1}, b_{k+1}$, TAG scale $\eta \geq 1$

1: Sample $\boldsymbol{x}_K \sim \mathcal{N}(\mathbf{0}, I)$
2: **for** $k = K-1, \ldots, 0$ **do**
3:      $\boldsymbol{\varepsilon}_{k+1} \leftarrow \epsilon_\theta(\boldsymbol{x}_{k+1}, t_{k+1})$                      ▷ noise prediction
4:      $\tilde{\boldsymbol{x}}_k \leftarrow a_{k+1}\boldsymbol{x}_{k+1} + b_{k+1}\,\boldsymbol{\varepsilon}_{k+1}$             ▷ e.g., `scheduler.step`
5:      $\Delta_{k+1} \leftarrow \tilde{\boldsymbol{x}}_k - \boldsymbol{x}_{k+1}$                         ▷ base increment
6:      $\widehat{\boldsymbol{x}}_{k+1} \leftarrow \boldsymbol{x}_{k+1}/\|\boldsymbol{x}_{k+1}\|_2$
7:      $\boldsymbol{P}_{\mathcal{M}_{k+1}} \leftarrow \widehat{\boldsymbol{x}}_{k+1}\widehat{\boldsymbol{x}}_{k+1}^\top, \quad \boldsymbol{P}_{\mathcal{M}_{k+1}}^\perp \leftarrow I - \boldsymbol{P}_{\mathcal{M}_{k+1}}$      ▷ projectors at $\boldsymbol{x}_{k+1}$
8:      $\boldsymbol{x}_k \leftarrow \boldsymbol{x}_{k+1} + \boldsymbol{P}_{\mathcal{M}_{k+1}}\Delta_{k+1} + \eta\,(\boldsymbol{P}_{\mathcal{M}_{k+1}}^\perp\Delta_{k+1})$      ▷ TAG amplification

---

via a local linearization:

$$\log p_\theta(\boldsymbol{x}) = \log p_\theta(\boldsymbol{x}_0) + (\boldsymbol{x} - \boldsymbol{x}_0)^\top \nabla_{\boldsymbol{x}} \log p_\theta(\boldsymbol{x})\big|_{\boldsymbol{x}=\boldsymbol{x}_0} + \mathcal{O}(\|\cdot\|^2). \tag{8}$$

Diffusion models (Song et al., 2020b; Ho et al., 2020) are designed to predict a score function, $\nabla_{\boldsymbol{x}} \log p(\boldsymbol{x} \mid t_k)\big|_{\boldsymbol{x}=\boldsymbol{x}_k} \approx -\epsilon_\theta(\boldsymbol{x}_k, t_k)/\sigma_k$, which operates on noisy versions of the data. Because diffusion models learn this score field, optimizing the global likelihood (equation 8) for a sample $\boldsymbol{x}_0$ during inference is *not directly tractable*. Therefore, we propose to apply the spirit of MLE at each *local step* of the sampling trajectory.

$$\log p(\boldsymbol{x}_k \mid t_{k+1}) \approx \log p(\boldsymbol{x}_{k+1} \mid t_{k+1}) + (\boldsymbol{x}_k - \boldsymbol{x}_{k+1})^\top \nabla_{\boldsymbol{x}} \log p(\boldsymbol{x} \mid t_{k+1})\big|_{\boldsymbol{x}=\boldsymbol{x}_{k+1}} + \mathcal{O}(\|\cdot\|^2). \tag{9}$$

The idea of enhancing a pre-trained score function with inference-time guidance has proven effective. For instance, when the score function is well trained on given training sets and this leads to well-trained maximum log-likelihood, we observe that the pre-trained score function could be improved by CFG (Ho & Salimans, 2021) which linearly biases the score toward the conditional target. Inspired by this, our approach provides inference-time guidance on the score function by maximizing the following local log-likelihood term, thereby guiding the sampling trajectory towards high-likelihood regions of the data distribution and reducing off-manifold artifacts (hallucination):

$$\max_{\boldsymbol{x}_k} (\boldsymbol{x}_k - \boldsymbol{x}_{k+1})^\top \nabla_{\boldsymbol{x}} \log p(\boldsymbol{x} \mid t_{k+1})\big|_{\boldsymbol{x}=\boldsymbol{x}_{k+1}} \tag{10}$$

**Single-step increment decomposition.** For deterministic DDIM/ODE samplers, the *single-step score state decomposition* can be written as

$$\Delta_{k+1} := \tilde{\boldsymbol{x}}_k - \boldsymbol{x}_{k+1} = \tilde{\alpha}_k \epsilon_\theta(\boldsymbol{x}_{k+1}, t_{k+1}) + \beta_k \boldsymbol{x}_{k+1}, \tag{11}$$

with coefficients

$$\tilde{\alpha}_k := \sigma_k - \frac{\sqrt{\bar{\alpha}_k}}{\sqrt{\bar{\alpha}_{k+1}}}\sigma_{k+1}, \quad \beta_k := \frac{\sqrt{\bar{\alpha}_k}}{\sqrt{\bar{\alpha}_{k+1}}} - 1, \quad \text{with} \quad \tilde{\alpha}_k < 0, \ \beta_k > 0,$$

where $\bar{\alpha}$ is the standard diffusion cumulative product term. Using the projection operators, which satisfy $\boldsymbol{P}_{\mathcal{M}_{k+1}}^\perp \boldsymbol{x}_{k+1} = 0$ and $\boldsymbol{P}_{\mathcal{M}_{k+1}}\boldsymbol{x}_{k+1} = \boldsymbol{x}_{k+1}$, yields the *projection-wise* identities

$$\boldsymbol{P}_{\mathcal{M}_{k+1}}^\perp \Delta_{k+1} = \tilde{\alpha}_k \boldsymbol{P}_{\mathcal{M}_{k+1}}^\perp \epsilon_\theta(\boldsymbol{x}_{k+1}, t_{k+1}), \quad \boldsymbol{P}_{\mathcal{M}_{k+1}}\Delta_{k+1} = \tilde{\alpha}_k \boldsymbol{P}_{\mathcal{M}_{k+1}}\epsilon_\theta(\boldsymbol{x}_{k+1}, t_{k+1}) + \beta_k \boldsymbol{x}_{k+1}. \tag{12}$$

Substituting equation 12 into the equation 7 gives

$$\boldsymbol{x}_k^{\text{TAG}} = \boldsymbol{x}_{k+1} + \tilde{\alpha}_k\big[\boldsymbol{P}_{\mathcal{M}_{k+1}} + \eta\boldsymbol{P}_{\mathcal{M}_{k+1}}^\perp\big]\epsilon_\theta(\boldsymbol{x}_{k+1}, t_{k+1}) + \beta_k \boldsymbol{x}_{k+1}, \quad \text{with} \quad \eta \geq 1. \tag{13}$$

Therefore, the TAG update $\Delta_{k+1}^{\text{TAG}}$ can be expressed in terms of the decomposed components of the original update $\Delta_{k+1}$:

$$\Delta_{k+1}^{\text{TAG}} = \big(\boldsymbol{P}_{\mathcal{M}_{k+1}} + \eta\,\boldsymbol{P}_{\mathcal{M}_{k+1}}^\perp\big)\Delta_{k+1}. \tag{14}$$

In this way, as visualized in Figure 2, semantic information can be isolated from the update vector via the tangential projection, thereby enabling semantics-aware amplification. To quantify its effect on the log-likelihood, assume the log-density is smooth (i.e., $\log p(\cdot|t_{k+1})$ is $C^2$ in a neighborhood of $\boldsymbol{x}_{k+1}$). The *first-order Taylor expansion gain* for a small TAG update $\Delta_{k+1}^{\text{TAG}} \in \mathbb{R}^d$ is

$$G(\eta) := \big(\Delta_{k+1}^{\text{TAG}}\big)^\top \nabla_{\boldsymbol{x}} \log p(\boldsymbol{x} \mid t_{k+1})\big|_{\boldsymbol{x}=\boldsymbol{x}_{k+1}}. \tag{15}$$

Next, we prove that increasing $\eta$ provides a monotonic increase in this first-order gain.

Table 1: **Quantitative results across previous guidance methods and +TAG sampling settings for unconditional generation.** Evaluated on the ImageNet *val* with 30K samples. All images are sampled with Stable Diffusion (SD) v1.5 (Rombach et al., 2022) using the DDIM sampler.

| Unconditional Generation | Guidance Scale | TAG Amp. ($\eta$) | #NFEs | #Steps | FID $\downarrow$ | IS $\uparrow$ |
|---|---|---|---|---|---|---|
| DDIM (Song et al., 2020a) | – | – | 50 | 50 | 76.942 | 14.792 |
| **TAG**$_{\text{SD v1.5}}$ | – | 1.05 | 50 | 50 | 67.971 | **16.620** |
| **TAG**$_{\text{SD v1.5}}$ | – | 1.15 | 50 | 50 | **67.805** | 16.487 |
| **TAG**$_{\text{SD v1.5}}$ | – | 1.25 | 50 | 50 | 71.801 | 15.815 |
| SAG (Hong et al., 2023) | 0.2 | – | 50 | 25 | 71.984 | 15.803 |
| **TAG + SAG** | 0.2 | 1.15 | 50 | 25 | **65.340** | **17.014** |
| PAG (Ahn et al., 2024a) | 3 | – | 50 | 25 | 64.595 | 19.30 |
| **TAG + PAG** | 3 | 1.15 | 50 | 25 | **63.619** | **19.90** |
| SEG (Hong, 2024) | 3 | – | 50 | 25 | 65.099 | 17.266 |
| **TAG + SEG** | 3 | 1.15 | 50 | 25 | **60.064** | **18.606** |

Table 2: **Quantitative comparison of *unconditional generation* on diffusion baselines.** We report results for ADM, Stable Diffusion (SD) v2.1, and SDXL. All models are evaluated on 10K ImageNet validation images using a DDIM sampler with 50 NFEs.

| Unconditional Generation | TAG Amp. ($\eta$) | #NFEs | #Steps | FID $\downarrow$ | IS $\uparrow$ |
|---|---|---|---|---|---|
| SD v2.1 (Rombach et al., 2022) | – | 50 | 50 | 100.977 | 11.553 |
| **TAG**$_{\text{SD v2.1}}$ | 1.15 | 50 | 50 | **88.788** | **13.311** |
| SDXL (Podell et al., 2024) | – | 50 | 50 | 124.407 | 9.034 |
| **TAG**$_{\text{SDXL}}$ | 1.20 | 50 | 50 | **113.798** | **9.716** |

Table 3: **Quantitative comparison of various ODE solvers on ImageNet.** We use Stable Diffusion (SD) v1.5 and compute all metrics over 30K samples. TAG achieves stronger performance *even with fewer #NFEs* for the DDIM sampler, and is also applicable to higher-order samplers such as DPM++. Inference time is measured using *torch.cuda.Event* and reported as the average over 100 consecutive runs on NVIDIA RTX 4090 GPUs.

| Unconditional Generation | TAG Amp. ($\eta$) | #NFEs | Inference Time (s) | FID $\downarrow$ | IS $\uparrow$ |
|---|---|---|---|---|---|
| DDIM (Song et al., 2020a) | – | 50 | 1.9507 | 76.942 | 14.792 |
| **TAG**$_{\text{DDIM}}$ | 1.15 | 25 | 1.0191 | 72.535 | 15.528 |
| **TAG**$_{\text{DDIM}}$ | 1.15 | 50 | 1.9674 | **67.805** | **16.487** |
| DPM++ 2S (Lu et al., 2025) | – | 10 | 0.4476 | 81.908 | 13.947 |
| **TAG**$_{\text{DPM++ 2S}}$ | 1.15 | 10 | 0.4501 | **77.654** | **15.026** |
| DPM++ 2M (Lu et al., 2025) | – | 10 | 0.4433 | 85.983 | 13.037 |
| **TAG**$_{\text{DPM++ 2M}}$ | 1.15 | 10 | 0.4522 | **74.238** | **14.930** |

**Theorem 4.1** (Monotonicity of the First-order Taylor Gain). *Assume a deterministic base step with* $\Delta_{k+1} = \tilde{\alpha}_k \epsilon_\theta(\boldsymbol{x}_{k+1}, t_{k+1}) + \beta_k \boldsymbol{x}_{k+1}$ *and* $\tilde{\alpha}_k \leq 0$. *Let* $\boldsymbol{P}_{\mathcal{M}_{k+1}} \succeq 0$ *and* $\boldsymbol{P}_{\mathcal{M}_{k+1}}^\perp \succeq 0$ *be the projectors defined above. For the TAG step* $\Delta_{k+1}^{\text{TAG}} = \boldsymbol{P}_{\mathcal{M}_{k+1}} \Delta_{k+1} + \eta \, \boldsymbol{P}_{\mathcal{M}_{k+1}}^\perp \Delta_{k+1}$, *the first-order Taylor gain* $G(\eta) := \left( \Delta_{k+1}^{\text{TAG}} \right)^\top \nabla_{\boldsymbol{x}} \log p(\boldsymbol{x} \mid t_{k+1}) \big|_{\boldsymbol{x}=\boldsymbol{x}_{k+1}}$ *satisfies*

$$\frac{\partial G(\eta)}{\partial \eta} \approx \frac{-\tilde{\alpha}_k}{\sigma_{k+1}} \left\| \boldsymbol{P}_{\mathcal{M}_{k+1}}^\perp \epsilon_\theta(\boldsymbol{x}_{k+1}, t_{k+1}) \right\|_2^2 \geq 0, \qquad (16)$$

*and, in particular,*

$$G^{\text{TAG}} - G^{\text{base}} = \underbrace{-\sigma_{k+1}^{-1} \cdot \left( \tilde{\alpha}_k(\eta - 1) \right)}_{\geq \, \mathbf{0} \; as \; \tilde{\alpha}_k \leq 0} \cdot \left\| \boldsymbol{P}_{\mathcal{M}_{k+1}}^\perp \epsilon_\theta(\boldsymbol{x}_{k+1}, t_{k+1}) \right\|_2^2 \geq 0, \qquad (17)$$

*Equality holds iff* $\eta = 1$. *The proof is provided in Appendix A.*

**Log-likelihood improvements via TAG.** We cast inference-time guidance as maximizing a log-likelihood gain (equation 10). TAG simply reweights the update step by amplifying the component that is orthogonal to the current state while leaving the parallel component unchanged. By Theorem 4.1, increasing the orthogonal weight monotonically raises the first-order Taylor gain, so TAG steers the sampler toward higher-density regions of the data manifold, improving image quality.

**Avoidance of normal amplification.** Amplifying the tangential component monotonically increases the first-order term of a Taylor gain of $\log p(\cdot \mid t_{k+1})$ (Theorem 4.1), which produces samples with less hallucination. However, amplifying the normal component increases radial contraction and leads to over-smoothing (Figure 5). This radial

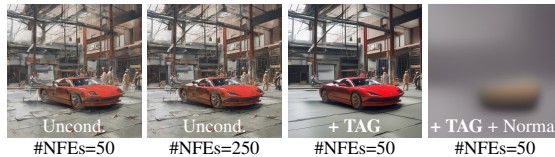

Figure 5: **Effectiveness of TAG**. At 50 NFEs, TAG surpasses the sample quality at 250 NFEs from baseline. In contrast, +*Normal* causes severe over-smoothing.

component of the single-step is aligned with the radial part of Tweedie's correction, which links $\boldsymbol{x}_k$ to the posterior mean $\mathbb{E}[\boldsymbol{x}_0|\boldsymbol{x}_k]$ via the score function (Tweedie et al., 1984; Song et al., 2020b). Formally, rescaling the normal part by a $\kappa \ (> 1)$, the radial first–order change is multiplied by $\kappa$:

$$\langle \widehat{\boldsymbol{x}}_{k+1}, \Delta_{k+1}^{(\kappa)} \rangle = \kappa \, \langle \widehat{\boldsymbol{x}}_{k+1}, \Delta_{k+1} \rangle. \tag{18}$$

Therefore, a value of $\kappa \ (> 1)$ excessively strengthens this contraction under the VP/DDIM schedule, leading to over-smoothing. In contrast, tangential scaling preserves the radial first–order term:

$$\langle \widehat{\boldsymbol{x}}_{k+1}, \Delta_{k+1}^{\text{TAG}} \rangle = \langle \widehat{\boldsymbol{x}}_{k+1}, \Delta_{k+1} \rangle. \tag{19}$$

To summarize, normal amplification breaks one–step calibration and *induces over-smoothing*, whereas tangential boosting improves alignment without disturbing the radial schedule.

## 4.2 Tangential Amplifying Guidance for Conditional Generation

Our analysis (§3, 4) shows that the *tangential* component encodes data-relevant directions and is radius-preserving to first-order; so amplifying it improves image quality by steering updates along data-aligned directions. In CFG (Ho & Salimans, 2021), the guided score combines conditional and unconditional branches

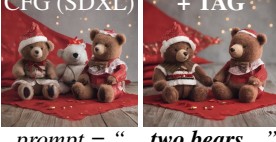

Figure 6: **CFG-based generation with SDXL.** Adding TAG improves text-condition faithfulness at matched NFEs.

$$\widetilde{\varepsilon}_k = \epsilon_\theta(\boldsymbol{x}_k, \boldsymbol{c}) + \omega(\epsilon_\theta(\boldsymbol{x}_k, \boldsymbol{c}) - \epsilon_\theta(\boldsymbol{x}_k, \emptyset)). \tag{20}$$

Because these two scores follow distinct trajectories, an incoherence between them can arise, and such an effect can degrade generation quality, an issue recently highlighted by Kwon et al. (2025). Motivated by this established score mismatch, and informed by our core intuition that the tangential field encodes data geometry (equation 1), we posit that this incoherence is fundamentally tangential in nature; that is, a persistent *mismatch* exists primarily between the conditional and unconditional tangential components.

**Conditional–unconditional tangent reconciliation.** Let $\boldsymbol{g}_k := \epsilon_\theta(\boldsymbol{x}_k, \boldsymbol{c}) - \epsilon_\theta(\boldsymbol{x}_k, \emptyset)$ denote the CFG guidance where $\epsilon_\theta(\cdot, \boldsymbol{c}), \epsilon_\theta(\cdot, \emptyset)$ denote the cond/unconditional predicted noise. We form a *conditional-relative tangent* by removing the unconditional tangent from the conditional one,

$$\boldsymbol{g}_k^\perp = \boldsymbol{P}_{\mathcal{M}}^\perp(\boldsymbol{x}_k)\big( \epsilon_\theta(\boldsymbol{x}_k, \boldsymbol{c}) \ - \ \epsilon_\theta(\boldsymbol{x}_k, \emptyset) \big) = \boldsymbol{P}_{\mathcal{M}}^\perp(\boldsymbol{x}_k)\boldsymbol{g}_k, \tag{21}$$

and *project* the conditional score $\epsilon_\theta(\boldsymbol{x}_k, \boldsymbol{c})$ onto this tangent subspace. We then amplify this condition relative tangent:

$$\tilde{\boldsymbol{\varepsilon}}_k \ = \ \epsilon_\theta(\boldsymbol{x}_k, \boldsymbol{c}) \ + \ \omega \boldsymbol{g}_k + \eta \left( \sigma_k^{-1} \boldsymbol{P}_{\mathcal{M}}(\boldsymbol{g}_k^\perp)\epsilon_\theta(\boldsymbol{x}_k, \boldsymbol{c}) \right), \tag{22}$$

where $\omega$ is the usual CFG scale and $\eta$ controls the extra tangential emphasis.

---

**Algorithm 2** Conditional TAG (C-TAG)

---

**Require:** Denoiser $\epsilon_\theta(\cdot)$, timesteps $\{t_k\}_{k=K}^0$, CFG scale $\omega$, TAG scale $\eta \geq 0$
1: Sample $\boldsymbol{x}_K \sim \mathcal{N}(\boldsymbol{0}, I)$ ▷ initialize from prior
2: **for** $k = K-1, \ldots, 0$ **do**
3:      $(\boldsymbol{\varepsilon}_u, \boldsymbol{\varepsilon}_c) \leftarrow \epsilon_\theta(\boldsymbol{x}_{k+1}, t_{k+1}, \cdot)$ ▷ uncond / cond noise
4:      $\boldsymbol{g}_k \leftarrow \boldsymbol{\varepsilon}_c - \boldsymbol{\varepsilon}_u$ ▷ CFG direction in $\boldsymbol{\varepsilon}$-space
5:      $\widehat{\boldsymbol{x}}_{k+1} \leftarrow \boldsymbol{x}_{k+1}/\|\boldsymbol{x}_{k+1}\|_2$
6:      $\boldsymbol{P}_{\mathcal{M}_{k+1}}^\perp \leftarrow I - \widehat{\boldsymbol{x}}_{k+1}\widehat{\boldsymbol{x}}_{k+1}^\top$ ▷ projector at $\boldsymbol{x}_{k+1}$
7:      $\boldsymbol{g}_k^\perp \leftarrow \boldsymbol{P}_{\mathcal{M}_{k+1}}^\perp \boldsymbol{g}_k$ ▷ tangential component
8:      $\tilde{\boldsymbol{\varepsilon}}_k \leftarrow \boldsymbol{\varepsilon}_u + \omega\boldsymbol{g}_k + \eta \left( \frac{\langle \boldsymbol{\varepsilon}_c, \boldsymbol{g}_k^\perp \rangle}{\|\boldsymbol{g}_k^\perp\|_2^2} \, \boldsymbol{g}_k^\perp \right)$ ▷ TAG-augmented CFG
9:      $\boldsymbol{x}_k \leftarrow \text{STEP}(\tilde{\boldsymbol{\varepsilon}}_k, t_{k+1}, \boldsymbol{x}_{k+1})$ ▷ scheduler step

---

Table 4: **Quantitative results across guidance-only (i.e. CFG, PAG, SEG) and guidance w/ TAG sampling settings.** Evaluated on the MS-COCO 2014 *val* split with 10k random text prompts. All images are sampled with Stable Diffusion v1.5 using the DDIM sampler. *cfg_scale=2.5*, *pag_scale=2.5* and *seg_scale=2.5* are applied for each experiments.[1]

| Conditional Generation | TAG Amp. ($\eta$) | #NFEs | #Steps | FID ↓ | CLIPScore ↑ |
|---|---|---|---|---|---|
| DDIM_cond-only | – | 30 | 30 | 85.145 | 19.77 ±3.43 |
| **TAG**_cond-only | 1.2 | 30 | 30 | **58.438** | **21.88** ±2.99 |
| CFG (Ho & Salimans, 2021) | – | 200 | 100 | 26.266 | 22.60 ± 3.28 |
| CFG_C-TAG | 2.5 | *60* | *30* | **23.414** | **22.82 ± 3.21** |
| CFG + PAG (Ahn et al., 2024a) | – | 50 | 25 | 24.280 | 22.72 ± 3.25 |
| CFG_C-TAG + PAG | 1.25 | 50 | 25 | **22.109** | 22.07 ± 3.49 |
| CFG + SEG (Hong, 2024) | – | 50 | 25 | 29.215 | **18.17 ± 3.55** |
| CFG_C-TAG + SEG | 1.25 | 50 | 25 | **23.446** | 16.94 ± 3.96 |

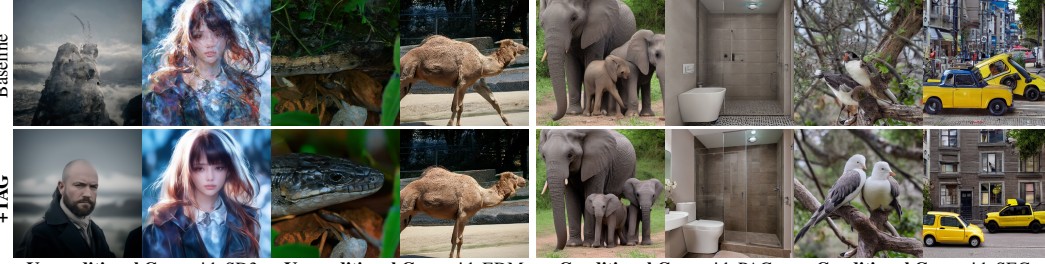

Unconditional Gen. with SD3    Unconditional Gen. with EDM    Conditional Gen. with PAG    Conditional Gen. with SEG

Figure 7: **Qualitative comparison of TAG across unconditional and conditional generation settings.** The left four columns demonstrate that for unconditional generation, TAG enhances the detail and coherence of samples from the SD3 (Podell et al., 2024) and EDM2 (Karras et al., 2024b). The right four columns show that for conditional generation, TAG can be applied on top of existing guidance methods (e.g., PAG (Ahn et al., 2024a), SEG (Hong, 2024)) to further improve their outputs.

## 5 EXPERIMENTS

**Backbones and inference setup.** We apply TAG at inference on pretrained backbones, using Stable Diffusion v1.5 (Rombach et al., 2022) for major experiments and Stable Diffusion 3 (Esser et al., 2024) for flow matching. Unconditional results are reported on ImageNet-1K *val* dataset (Deng et al., 2009). Text-conditional results use MS-COCO 2014 *val* dataset (Lin et al., 2015). The number of function evaluations (#NFEs) follows each table. TAG is inserted after every solver update with amplification $\eta$. Metrics include FID (Heusel et al., 2017), IS (Salimans et al., 2016), CLIPScore (Hessel et al., 2021), FD-dino (Stein et al., 2023) and NFEs. FID is computed with *pytorch-fid* (Seitzer, 2020), IS with Inception-V3 (Szegedy et al., 2016), and CLIPScore is computed with OpenAI CLIP ViT-L/14. All runs use fixed seeds and identical evaluation to the corresponding baselines.

**Improvements on conditional generation.** Table 4 presents quantitative results on the MS-COCO, demonstrating that augmenting existing guidance samplers with TAG consistently yields substantial improvements in sample fidelity while largely preserving text-image alignment. Notably, TAG enables a 30 steps sampling process to outperform the 100 steps CFG baseline. Even in a condition only setting, TAG dramatically reduces FID and increases CLIPScore, confirming its *foundational benefits independent of a guidance signal*. Furthermore, this

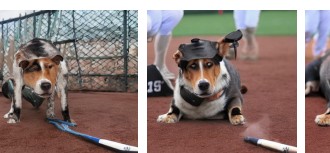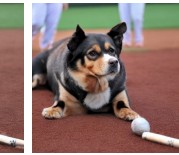

$\omega = 2.5, \eta = 0.0$  $\omega = 5.0, \eta = 0.0$  $\omega = 2.5, \eta = 1.0$

Figure 8: **Qualitative Results with CFG.** TAG produces higher-fidelity samples with fewer hallucinations, outperforming even baselines with a higher CFG scale $\omega$.

trend extends to other guidance techniques such as PAG and SEG, where TAG again reduces FID at the same computational cost. The qualitative improvements are visualized in Figure 8, 14 and Figure 15, which demonstrates TAG's ability to produce higher-fidelity images with fewer artifacts.

---

[1]This table highlights fundamental efficacy in controlled regimes; for extended evaluations in practical high-fidelity regimes, see Appendix §C.

Table 5: **Quantitative results for geometry-aware guidance (APG, TCFG) and TAG.** Evaluated on MS-COCO 2014 *val* with 10k random prompts. All images are sampled with SDXL using 50 steps and a fixed CFG scale of 5.0. Additional qualitative results are provided in Appendix §F.1.

| Cond. Gen. | scale ($\eta$) | $[\eta_{\mathrm{sta}}, \eta_{\mathrm{end}}]$ | FID ↓ | CLIPScore ↑ |
|---|---|---|---|---|
| CFG | – | – | 19.705 | $26.30 \pm 3.33$ |
| + TCFG | – | – | 20.719 | $25.63 \pm 3.38$ |
| + APG | – | – | 19.523 | $\mathbf{26.71 \pm 3.31}$ |
| CFG$_{\text{C-TAG}}$ | 0.35 | [1000, 0] | 19.798 | $26.30 \pm 3.60$ |
| CFG$_{\text{C-TAG}}$ | 0.35 | [1000, 800] | **19.288** | $26.23 \pm 3.47$ |

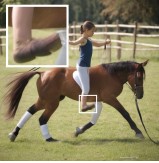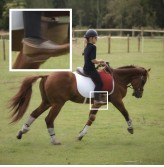

APG         CFG$_{\text{C-TAG}}$

**Improvements on unconditional generation.** For unconditional generation, TAG consistently improves sample quality across a range of models and samplers. As shown in Table 1, it reduces FID and increases IS at a matched NFEs. Notably, TAG acts as a 'plug-and-play' module for existing guidance methods (e.g., SAG, PAG, SEG), enhancing their performance without architectural changes or additional model evaluations. Moreover, TAG significantly pushes the compute–quality frontier by enabling both faster inference and higher quality. With samplers like DDIM and DPM++, TAG can achieve superior results with as few as half the NFEs (Table 3). Concurrently, it substantially boosts performance on foundational models like SD v2.1 and SDXL at a fixed computational cost (Table 2). This dual benefit provides a practical path to faster inference and extends to SOTA models like SD3 (Table 6), with qualitative improvements visualized in Figures 7 and 13.

**Improvements on Flow Matching.** By consistently guiding the sampling trajectories toward regions of high probability, TAG serves as a broadly applicable enhancement for generative ODE solvers, whether the underlying training scheme is score-based or flow-matching. Figure 7

Table 6: **Quantitative results** for Stable Diffusion 3. Evaluations are conducted on ImageNet *val* with 30K samples; all images are generated with 50 NFEs.

| Unconditional Generation | scale ($\eta$) | FID ↓ | IS ↑ |
|---|---|---|---|
| SD3 (Esser et al., 2024) | – | 96.383 | 11.831 |
| **TAG$_{\text{SD3}}$** | 1.05 | **91.706** | **12.274** |

and Table 6 indicate that TAG transfers to flow-matching backbones (Esser et al., 2024). Inserted as a lightweight tangential reweighting after each solver step, without architectural changes or additional function evaluations. TAG yields a modest but consistent FID improvement at matched compute and visibly reduces artifacts in unconditional samples. These results show TAG's potential to be model-agnostic across diverse architectures, including modern large-scale models.

**Improvements on Modern Diffusion Backbone.** To assess whether TAG remains effective on a strong modern backbone, we apply it to EDM2 (Karras et al., 2024b) trained on ImageNet-512. As summarized in Table 7, TAG consistently improves the semantic quality of generated samples, as measured by FD-DINOv2 (Stein et al., 2023), while largely preserving FID. A mild configuration

Table 7: **TAG ablation on EDM2.** We report FID and FD-DINOv2 for the EDM2. All models are evaluated with official generation-evaluation process with 10k samples. $[\eta_{\mathrm{sta}}, \eta_{\mathrm{end}}]$ denotes the range of diffusion steps where TAG is applied.

| Uncond. Gen. | $[\eta_{\mathrm{sta}}, \eta_{\mathrm{end}}]$ | scale ($\eta$) | FID ↓ | FD-dino-v2 ↓ |
|---|---|---|---|---|
| EDM2 (Karras et al., 2024b) | – | – | 5.185 | 100.708 |
| **TAG$_{\text{EDM2}}$** | [0, 150] | **1.12** | **5.164** | 99.211 |
| **TAG$_{\text{EDM2}}$** | [0, 150] | 1.18 | 5.204 | 98.567 |
| **TAG$_{\text{EDM2}}$** | [0, 150] | 1.25 | 5.228 | **98.148** |
| **TAG$_{\text{EDM2}}$** | [0, 600] | 1.07 | 5.630 | 95.883 |

slightly *improves* both FD-DINOv2 and FID, demonstrating that TAG can be added to EDM2. Increasing the scale further yields larger semantic gains at the cost of a modest FID degradation, and applying TAG over a longer window provides even stronger semantic improvements (Figure 13).

**Comparison with geometry-aware guidance.** Table 5 and §F.1 compare C-TAG with recent geometry-aware guidance methods (Sadat et al., 2025; Kwon et al., 2025). Under the default SDXL setting, C-TAG is comparable but can be slightly behind APG. However, as discussed in §6, introducing a *windowed schedule* improves the standalone performance of C-TAG, predominantly by affecting early denoising where global layout is determined while avoiding overly aggressive constraints during later refinement steps. Although this improvement can coincide with a modest decrease in the *average* CLIPScore, the generations still sufficiently reflect the prompts, as shown in §F.1. Finally, we emphasize that TAG is a sampler-level modification of the solver update rather than a change to the guidance algebra, and is therefore broadly applicable beyond the SDXL-CFG setting considered here, including CFG-free (Table. 4, **TAG$_{\text{cond-only}}$**) and unguided sampling regimes.

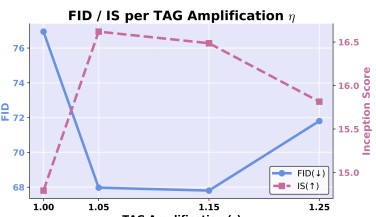
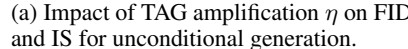

(a) Impact of TAG amplification $\eta$ on FID and IS for unconditional generation.

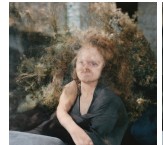
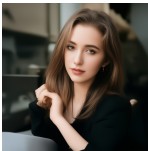
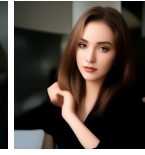
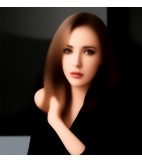

| $\eta = 1.0$ | $\eta = 1.1$ | $\eta = 1.2$ | $\eta = 1.3$ |

(b) Qualitative comparison across amplification levels $\eta$: moderate tangential amplification enhances detail and coherence, while excessive amplification degrades fidelity.

Figure 9: **Ablation on TAG amplification $\eta$.** Figure 9a and Table 1 show gains at moderate $\eta$ and degradation when amplification is *excessive*. Figure 9b confirms the same trend for Flow-matching, underscoring the need to select an appropriate $\eta$.

**Image-to-3D Shape Generation.** To investigate the applicability of TAG beyond standard 2D image synthesis, we conducted a preliminary qualitative experiment on Image-to-3D generation using SV3D (Voleti et al., 2024). As shown in Figure 10, we applied TAG during the sampling process of novel view synthesis. Compared to the baseline, the addition of TAG yields modest yet noticeable improvements in preserving semantic details relative to the Ground Truth. These results suggest that TAG's mechanism of refining the sampling trajectory has potential utility in higher-dimensional generative tasks.

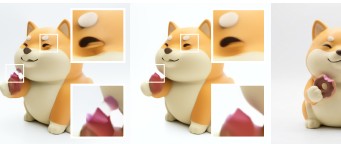

SV3D  + TAG  GT

Figure 10: **Qualitative comparison** on SV3D (Voleti et al., 2024). Adding TAG yields modest gains in semantic consistency with the GT, suggesting potential applicability to broader application.

## 6 LIMITATION & DISCUSSION

An ablation of $\eta$ (Fig. 9a, Tab. 1; see also Fig. 9b for flow matching) shows that while moderate tangential amplification improves quality, excessive $\eta$ degrades it. Analytically, the post-step state norm satisfies:

Table 8: **Effect of timestep windowing in TAG.** TAG is activated only during a timestep window $[\eta_{\text{sta}}, \eta_{\text{end}}]$, motivated by the Gaussian annulus theorem. We evaluate FID/IS using 30K ImageNet *val* samples with Stable Diffusion v1.5, DDIM sampler (Song et al., 2020a).

| Uncond. | $[\eta_{\text{sta}}, \eta_{\text{end}}]$ | scale ($\eta$) | #NFEs | #Steps | FID $\downarrow$ | IS $\uparrow$ |
|---|---|---|---|---|---|---|
| **TAG**$_{\text{SD v1.5}}$ | [400, 0] | 1.15 | 50 | 50 | 69.428 | 16.104 |
| **TAG**$_{\text{SD v1.5}}$ | [1000,0] | 1.15 | 50 | 50 | 67.805 | 16.487 |
| **TAG**$_{\text{SD v1.5}}$ | [1000,400] | 1.15 | 50 | 50 | **63.870** | **17.516** |

$$\|\boldsymbol{x}_{k+1} + \Delta_{k+1}^{\text{TAG}}\|_2^2 = \|\boldsymbol{x}_{k+1} + \Delta_{k+1}\|_2^2 + \overbrace{(\eta^2 - 1)}^{\text{additive term}} \|\boldsymbol{P}_{\mathcal{M}_{k+1}}^{\perp} \Delta_{k+1}\|_2^2. \tag{23}$$

For large $\eta$, the second term perturbs the scheduler's calibration, causing the observed degradation.

Furthermore, while our decomposition is motivated by the *Gaussian annulus theorem* (Blum et al., 2020), the geometric landscape evolves during sampling. As latents approach the clean data manifold, the diffusion score becomes dominated by the component in the *normal bundle* (Stanczuk et al., 2023). Consequently, as shown in the Figure 2, the tangential component loses informativeness in the small-noise regime, and amplifying it may introduce artifacts. Validating this (Table 8), we find that a *windowed* application of TAG outperforms the full-trajectory approach. This aligns with findings that the generation trajectory is largely predetermined by the initialization (Ahn et al., 2024b), confirming that TAG is most effective before the manifold convergence phase.

## 7 CONCLUSION

This paper introduces a new perspective for addressing the problem of hallucinations in diffusion models, demonstrating that the tangential component of the sampling update encodes critical semantic structure. Based on this geometric insight, we propose **T**angential **A**mplifying **G**uidance **(TAG)**, a practical, architecture-agnostic method that amplifies the tangential component. By doing so, TAG effectively steers the sampling trajectory toward higher-density regions of the data manifold, generating samples with fewer hallucinations and improved fidelity. Our method achieved good samples without requiring retraining or incurring any additional heavy computational overhead, offering a practical, plug-and-play solution for enhancing existing diffusion model backbones.

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

# APPENDIX

| Symbol | Meaning |
|---|---|
| $\boldsymbol{x}_k \in \mathbb{R}^d$ | latent at step $k$ (time $t_k$). |
| $\widehat{\boldsymbol{x}} := \boldsymbol{x}/\|\boldsymbol{x}\|_2$ | Unit vector in the direction of $\boldsymbol{x}$. |
| $\mathcal{M}$ | Noisy data manifold (iso-noise surface) |
| $\boldsymbol{P}_{\mathcal{M}_k} := \widehat{\boldsymbol{x}}_k \widehat{\boldsymbol{x}}_k^\top$ | Projector onto $\mathrm{span}(\boldsymbol{x}_k)$ ('normal' to the iso-noise surface at $\boldsymbol{x}$). |
| $\boldsymbol{P}_{\mathcal{M}_k}^\perp := I - \boldsymbol{P}_{\mathcal{M}_k}$ | Tangential projector orthogonal to $\boldsymbol{x}_k$. |
| $\langle \boldsymbol{u}, \boldsymbol{v} \rangle, \|\boldsymbol{u}\|_2$ | Euclidean inner product and norm. |
| $\boldsymbol{s}_\theta(x, t)$ | Model score. |
| $\epsilon_\theta(x, t)$ | Model noise prediction; $\boldsymbol{s}_\theta(x, t) = -\epsilon_\theta(x, t)/\sigma_t$. |
| $\boldsymbol{g}_k$ | Guidance residual direction. |
| $\Delta_k$ | Base solver increment without TAG at step $k$. |
| $\Delta_k^{\mathrm{TAG}}$ | TAG-modified increment at step $k$. |
| $\eta \geq 1$ | TAG tangential amplification factor (scales $\boldsymbol{P}_k^\perp \Delta$). |
| NFEs | Number of function evaluations. |

# A  PROOF & DERIVATION

**Proof for Theorem 4.1**

*Proof.* Assume the deterministic base step $\Delta_{k+1} = \tilde{\alpha}_k \epsilon_\theta(\boldsymbol{x}_{k+1}, t_{k+1}) + \beta_k \boldsymbol{x}_{k+1}$, with $\tilde{\alpha}_k \leq 0$, and let $\boldsymbol{P}_{\mathcal{M}_{k+1}}, \boldsymbol{P}_{\mathcal{M}_{k+1}}^\perp$ be the orthogonal projectors with $\boldsymbol{P}_{\mathcal{M}_{k+1}} \boldsymbol{x}_{k+1} = \boldsymbol{x}_{k+1}$ and $\boldsymbol{P}_{\mathcal{M}_{k+1}}^\perp \boldsymbol{x}_{k+1} = \boldsymbol{0}$. Applying the projectors to the base decomposition gives

$$\boldsymbol{P}_{\mathcal{M}_{k+1}}^\perp \Delta_{k+1} = \tilde{\alpha}_k \, \boldsymbol{P}_{\mathcal{M}_{k+1}}^\perp \epsilon_\theta(\boldsymbol{x}_{k+1}, t_{k+1}), \tag{24}$$

$$\boldsymbol{P}_{\mathcal{M}_{k+1}} \Delta_{k+1} = \tilde{\alpha}_k \, \boldsymbol{P}_{\mathcal{M}_{k+1}} \epsilon_\theta(\boldsymbol{x}_{k+1}, t_{k+1}) + \beta_k \, \boldsymbol{x}_{k+1}. \tag{25}$$

Therefore, the TAG update rule step is

$$\Delta_{k+1}^{\mathrm{TAG}} = \big( \boldsymbol{P}_{\mathcal{M}_{k+1}} + \eta \, \boldsymbol{P}_{\mathcal{M}_{k+1}}^\perp \big) \Delta_{k+1} = \tilde{\alpha}_k \big[ \boldsymbol{P}_{\mathcal{M}_{k+1}} + \eta \boldsymbol{P}_{\mathcal{M}_{k+1}}^\perp \big] \epsilon_\theta(\boldsymbol{x}_{k+1}, t_{k+1}) + \beta_k \boldsymbol{x}_{k+1}. \tag{26}$$

The first-order Taylor gain with respect to TAG update at $t_{k+1}$ is defined as:

$$G(\eta) := \big( \Delta_{k+1}^{\mathrm{TAG}} \big)^\top \nabla_{\boldsymbol{x}} \log p(\boldsymbol{x} \mid t_{k+1}) \big|_{\boldsymbol{x} = \boldsymbol{x}_{k+1}}$$

$$= \Big( \big( \boldsymbol{P}_{\mathcal{M}_{k+1}} + \eta \, \boldsymbol{P}_{\mathcal{M}_{k+1}}^\perp \big) \Delta_{k+1} \Big)^\top \nabla_{\boldsymbol{x}} \log p(\boldsymbol{x} \mid t_{k+1}) \big|_{\boldsymbol{x} = \boldsymbol{x}_{k+1}} \tag{27}$$

We analyze this gain by approximating the true score with the model's score function

$$\boldsymbol{s}_\theta(\boldsymbol{x}_{k+1}, t_{k+1}) = -\sigma_{k+1}^{-1} \epsilon_\theta(\boldsymbol{x}_{k+1}, t_{k+1}), \tag{28}$$

thus:

$$G(\eta) = \Big( \big( \boldsymbol{P}_{\mathcal{M}_{k+1}} + \eta \, \boldsymbol{P}_{\mathcal{M}_{k+1}}^\perp \big) \Delta_{k+1} \Big)^\top \nabla_{\boldsymbol{x}} \log p(\boldsymbol{x} \mid t_{k+1}) \big|_{\boldsymbol{x} = \boldsymbol{x}_{k+1}}$$

$$\approx \Big( \big( \boldsymbol{P}_{\mathcal{M}_{k+1}} + \eta \, \boldsymbol{P}_{\mathcal{M}_{k+1}}^\perp \big) \Delta_{k+1} \Big)^\top \boldsymbol{s}_\theta(\boldsymbol{x}_{k+1}, t_{k+1})$$

$$= -\sigma_{k+1}^{-1} \cdot \Big( \big( \boldsymbol{P}_{\mathcal{M}_{k+1}} + \eta \, \boldsymbol{P}_{\mathcal{M}_{k+1}}^\perp \big) \Delta_{k+1} \Big)^\top \epsilon_\theta(\boldsymbol{x}_{k+1}, t_{k+1}) \tag{29}$$

Substitute equation 26 into equation 29, then:

$$G(\eta) \approx -\sigma_{k+1}^{-1} \Big( \tilde{\alpha}_k \boldsymbol{P}_{\mathcal{M}_{k+1}} \epsilon_\theta + \beta_k \boldsymbol{x}_{k+1} + \eta \tilde{\alpha}_k \boldsymbol{P}_{\mathcal{M}_{k+1}}^\perp \epsilon_\theta \Big)^\top \epsilon_\theta$$

$$= -\sigma_{k+1}^{-1} \Big( \tilde{\alpha}_k (\boldsymbol{P}_{\mathcal{M}_{k+1}} \epsilon_\theta)^\top \epsilon_\theta + \beta_k \boldsymbol{x}_{k+1}^\top \epsilon_\theta + \eta \tilde{\alpha}_k (\boldsymbol{P}_{\mathcal{M}_{k+1}}^\perp \epsilon_\theta)^\top \epsilon_\theta \Big). \tag{30}$$

Since $\boldsymbol{P}$ and $\boldsymbol{P}^\perp$ are symmetric and idempotent, thus

$$\boldsymbol{v}^\top \boldsymbol{P} \boldsymbol{v} = \|\boldsymbol{P} \boldsymbol{v}\|_2^2 \tag{31}$$

is established. Therefore,

$$G(\eta) \approx -\sigma_{k+1}^{-1}\left(\tilde{\alpha}_k\big\|\boldsymbol{P}_{\mathcal{M}_{k+1}}\epsilon_\theta\big\|_2^2 + \beta_k\boldsymbol{x}_{k+1}^\top\epsilon_\theta + \eta\tilde{\alpha}_k\big\|\boldsymbol{P}_{\mathcal{M}_{k+1}}^\perp\epsilon_\theta\big\|_2^2\right). \tag{32}$$

Differentiating the gain $G(\eta)$ in equation 32 with respect to $\eta$ yields:

$$\frac{\partial G(\eta)}{\partial \eta} \approx \frac{-\tilde{\alpha}_k}{\sigma_{k+1}}\left|\boldsymbol{P}_{\mathcal{M}_{k+1}}^\perp\epsilon_\theta(\boldsymbol{x}_{k+1}, t_{k+1})\right|_2^2 \geq 0. \tag{33}$$

This derivative is guaranteed to be non-negative, since the DDIM sampler coefficient $\tilde{\alpha}_k \leq 0$ by definition, while $\sigma_{k+1}$ and the squared L2-norm are strictly non-negative. This proves that the first-order gain $G(\eta)$ is a monotonically non-decreasing function of $\eta$. Consequently, amplifying the tangential component of the update step via TAG is guaranteed to improve the first-order log-likelihood gain compared to the base update step.

**Analysis on pure TAG gain.** Subtracting each gain $G^{\text{base}} \triangleq G(\eta = 1)$ and $G^{\text{TAG}} \triangleq G(\eta > 1)$,

$$\overbrace{\left(-\sigma_{k+1}^{-1}\cdot\left(\Delta_{k+1}^{\text{TAG}}\right)^\top\epsilon_\theta(\boldsymbol{x}_{k+1}, t_{k+1})\right)}^{\text{TAG update gain, }G^{\text{TAG}}} - \overbrace{\left(-\sigma_{k+1}^{-1}\cdot\left(\Delta_{k+1}\right)^\top\epsilon_\theta(\boldsymbol{x}_{k+1}, t_{k+1})\right)}^{\text{base update gain, }G^{\text{base}}}$$

$$= -\sigma_{k+1}^{-1}\cdot\left(\Delta_{k+1}^{\text{TAG}} - \Delta_{k+1}\right)^\top\epsilon_\theta(\boldsymbol{x}_{k+1}, t_{k+1})$$

$$= -\sigma_{k+1}^{-1}\cdot\left((\eta-1)\,\boldsymbol{P}_{\mathcal{M}_{k+1}}^\perp\Delta_{k+1}\right)^\top\epsilon_\theta(\boldsymbol{x}_{k+1}, t_{k+1}). \tag{34}$$

Using $\Delta_{k+1} = \tilde{\alpha}_k\,\epsilon_\theta(\boldsymbol{x}_{k+1}, t_{k+1}) + \beta_k\,\boldsymbol{x}_{k+1}$, $\boldsymbol{P}_{\mathcal{M}_{k+1}}^\perp$ be:

$$\boldsymbol{P}_{\mathcal{M}_{k+1}}^\perp\Delta_{k+1} = \boldsymbol{P}_{\mathcal{M}_{k+1}}^\perp\tilde{\alpha}_k\,\epsilon_\theta(\boldsymbol{x}_{k+1}, t_{k+1}). \tag{35}$$

Thus, substitute equation 35 into equation 34 then:

$$G^{\text{TAG}} - G^{\text{base}} = \underbrace{-\sigma_{k+1}^{-1}\cdot\left(\tilde{\alpha}_k(\eta-1)\right)}_{\text{scalar}} \cdot \left(\boldsymbol{P}_{\mathcal{M}_{k+1}}^\perp\,\epsilon_\theta(\boldsymbol{x}_{k+1}, t_{k+1})\right)^\top\epsilon_\theta(\boldsymbol{x}_{k+1}, t_{k+1}). \tag{36}$$

This simplifies to the final quadratic form:

$$G^{\text{TAG}} - G^{\text{base}} = \underbrace{-\sigma_{k+1}^{-1}\cdot\left(\tilde{\alpha}_k(\eta-1)\right)}_{\geq\,\boldsymbol{0}\ \text{as}\ \tilde{\alpha}_k\,\leq\,0} \cdot \big\|\boldsymbol{P}_{\mathcal{M}_{k+1}}^\perp\epsilon_\theta(\boldsymbol{x}_{k+1}, t_{k+1})\big\|_2^2, \tag{37}$$

This proves that the difference in gain is non-negative for any $\eta \geq 1$. Therefore, the first-order log-likelihood gain of the TAG update is always greater than or equal to that of the base update, with equality holding if and only if $\eta = 1$ or the tangential component of the score is zero. $\qquad\square$

# B DISCUSSION ON ORTHOGONAL AND PROJECTION-BASED GUIDANCE

Several recent works incorporate projections or tangential/orthogonal notions in diffusion guidance. While they share surface-level similarity, they differ in (i) *which quantity is decomposed*, (ii) how the relevant *subspace is defined*, and (iii) the *problem setting* and objective that motivate the projection. We summarize these distinctions below to situate TAG.

## B.1 ORTHOGONAL AND PROJECTION-BASED DIFFUSION GUIDANCE

### B.1.1 CFG-SPECIFIC PROJECTIONS FOR ARTIFACT SUPPRESSION

**Projected CFG for oversaturation.** Sadat et al. (2025) analyze artifacts such as oversaturation at large CFG (Ho & Salimans, 2021) scales. They decompose the CFG update into components parallel and orthogonal to the *conditional* score, and empirically identify the *parallel component as the primary cause of saturation*.

Their Adaptive Projected Guidance (APG) therefore *attenuates the parallel component* while preserving the orthogonal component, and complements this with rescaling and momentum-style heuristics motivated by a gradient-ascent interpretation of CFG. This line of work is **(i) tied to the CFG algebra** and targets **(ii) large-scale saturation** artifacts.

**SVD-based tangential damping within CFG.** Kwon et al. (2025) address conditional-unconditional misalignment in CFG, which can lead to off-manifold samples. They form a score matrix from conditional and unconditional scores and perform SVD, interpreting high singular-value directions as shared, approximately manifold-normal components and **low singular-value directions as tangential components**.

They observe that the discrepancy between conditional and unconditional scores is concentrated in low-SV (i.e., tangential) directions, and propose filtering the *unconditional* score by **projecting it onto the shared high-SV subspace** before applying CFG. This approach modifies only the unconditional term inside CFG rather than the base solver update, and does not study the semantic role of tangential components along the solver trajectory.

### B.1.2 CORRECTION OF CFG DYNAMICS

**Nonlinear correction via characteristics.** Zheng & Lan (2024) derive guidance from the nonlinear Fokker-Planck dynamics of guided diffusion. They show that standard linear *CFG neglects nonlinear correction terms* that become important at high guidance scales, causing artifacts.

Their method solves a nonlinear *fixed-point / characteristic equation* for the corrected guided score. A **projection operator** can be inserted for numerical regularization, but it is not the conceptual driver of the method; the theoretically **ideal operator is the identity**. Thus, improvements come from nonlinear correction *rather than from projecting tangential/orthogonal components*.

### B.1.3 PROMPT INTERACTION AND SEMANTIC DISENTANGLEMENT

**Orthogonalizing negative prompts.** Armandpour et al. (2023) study failures of negative prompting when negative and positive concepts overlap. They decompose the negative score into components **parallel and perpendicular to the positive score** direction, discard the parallel (overlapping) negative component, and apply only the perpendicular negative guidance.

The goal is to prevent negative prompts from canceling shared desirable attributes, especially in text-to-image and text-to-3D pipelines. This mechanism concerns *prompt interaction effects* rather than diffusion-step geometry.

### B.1.4 TASK-SPECIFIC MANIFOLD CONSTRAINTS

**Tangent restriction for loss-based guidance.** He et al. (2023) focus on training-free, loss-based guidance (e.g., **DPS (Chung et al., 2024)-style inverse problems**), noting that ambient-space loss gradients may *violate data-manifold* constraints.

Table 9: **Comparison of projection / tangential guidance methods.** Prior works decompose CFG algebra, prompt gradients, or task-specific manifold scores; TAG uniquely decomposes and amplifies the *single-step solver update* w.r.t. iso-noise manifolds, without auxiliary models.

| Work | Decomp. quantity | Subspace / manifold | Projection / filtering | Goal / setting | Cost |
|---|---|---|---|---|---|
| APG (Sadat et al., 2025) | CFG update | $\parallel$ / $\perp$ to conditional score | Downweight $\parallel$ part in CFG | Fix high-scale CFG oversaturation | CFG |
| TCFG (Kwon et al., 2025) | Cond.+uncond. scores | SVD: high-SV shared vs. low-SV tangential | Project *uncond.* $\rightarrow$ high-SV before CFG | Reduce cond/uncond mismatch (T2I) | CFG |
| Characteristic Guidance (Zheng & Lan, 2024) | Guided score fixed-point | None (proj. optional) | Optional stabilizing $P$; ideal $P=I$ | Nonlinear CFG correction | CFG |
| Perp-Neg (Armandpour et al., 2023) | Neg-prompt score | $\parallel$ / $\perp$ to pos. prompt | Remove $\parallel$ neg. component | Avoid neg/pos overlap (T2I/T2–3D) | Multi-prompt |
| MPGD (He et al., 2023) | Loss grad on $\boldsymbol{x}_{0\|t}$/latent | tangent of AE-driven $\mathcal{M}_0$ | Project loss grad $\rightarrow$ tangent | On-manifold inverse problems | AE extra |
| SDDM (Sun et al., 2023) | I2I cond. score/energy | Ref-induced $\mathcal{M}_t(y_0)$ | Normal transport vs. tangential refine | Unpaired I2I, task-specific | Task mods |
| TAG (ours) | Solver step $\triangle\boldsymbol{x}_k$ | $\parallel$ / $\perp$ to iso-noise manifold $\mathcal{M}_k$ | Amplify tangential of $\triangle\boldsymbol{x}_k$ | Generic sampling refinement | simple vector calculation §D |

They apply **guidance on $\boldsymbol{x}_{0|t}$** and project external guidance gradients onto tangent spaces of the clean-data manifold $\mathcal{M}_0$, estimating tangents using an auxiliary pretrained autoencoder. These methods rely on an explicit manifold model and are tailored to **observation-conditioned tasks** rather than generic unconditional or standard conditional generation.

**Refinement–transport decomposition in conditional I2I.** In unpaired conditional image-to-image translation, Sun et al. (2023) construct **reference-dependent manifolds $\mathcal{M}_t(y_0)$** induced by a reference image $y_0$. Such a conditional structure is crucial: the authors emphasize (§4, Lemma 1 of (Sun et al., 2023)) that in standard diffusion, where intermediate manifolds at adjacent timesteps are coupled, a **tangential/normal score split is generally meaningless**.

In contrast, Sun et al. (2023) argue that the **I2I setting** yields compact, well-separated manifolds across time, making the **decomposition meaningful**; under this specific condition, the tangential component acts as an on-manifold refinement term, while the normal component governs transport between manifolds of adjacent timesteps.

### B.2 TAG: Intrinsic Tangential Amplification for General Diffusion Sampling

TAG leverages the *intrinsic geometry* of diffusion trajectories by decomposing the *single-step solver update* at each state $\boldsymbol{x}_k$ into radial (manifold-normal) and tangential components with respect to the iso-noise manifold $\mathcal{M}_k$. We show that this **split is stable and meaningful** in general diffusion:

- the tangential update captures semantic refinement along $\mathcal{M}_k$
- while the radial part is largely unstructured,

and amplifying the tangential component provably promotes local log-likelihood ascent. In contrast to CFG-algebraic projections (Sadat et al., 2025; Kwon et al., 2025), nonlinear CFG corrections (Zheng & Lan, 2024), prompt-specific orthogonalization (Armandpour et al., 2023), or task-/condition-dependent manifold models (He et al., 2023; Sun et al., 2023), TAG operates directly on the base solver update, requires *no auxiliary models* or *extra network passes*, and *applies uniformly* to unconditional and standard conditional generation across modern backbones.

## C ANALYSIS IN PRACTICAL HIGH-FIDELITY REGIMES

Table 10: **Quantitative results across conditional generation.** Evaluated on the MS-COCO 2014 *val* split with 20k random text prompts. All images are sampled with Stable Diffusion v2.1 using the DDIM sampler. *cfg_scale=7.5* is applied for each experiments.

| Conditional Generation | scale ($\eta$) | #Steps | FID ↓ | CLIPScore ↑ | FD-dino-v2 ↓ | ImageReward ↑ |
|---|---|---|---|---|---|---|
| CFG (Ho & Salimans, 2021) | – | 50 | **18.842** | 26.43 ± 3.20 | **236.34** | 0.102 |
| CFG$_{\text{C-TAG}}$ | 0.05 | 50 | 19.177 | 26.45 ± 3.21 | 239.73 | 0.112 |
| CFG$_{\text{C-TAG}}$ | 0.10 | 50 | 19.548 | 26.46 ± 3.23 | 242.24 | 0.117 |
| CFG$_{\text{C-TAG}}$ | 0.15 | 50 | 19.884(+5.5%) | **26.48 ± 3.24** | 245.23(+3.7%) | **0.122**(+19.6%) |

Table 11: **Paired improvements on MS-COCO2014 val (20k samples).** We report mean paired gains over the baseline with bootstrap 95% CIs and win-rate (higher-is-better).

| Method | $\eta$ | Steps | Δ CLIPScore (95% CI) | Win(%) | Δ ImageReward (95% CI) | Win(%) |
|---|---|---|---|---|---|---|
| CFG | – | 50 | – | – | – | – |
| CFG$_{\text{C-TAG}}$ | 0.05 | 50 | +0.0178 [−0.0017, +0.0364] | 50.9 | +0.0103 [+0.0052, +0.0154] | 51.3 |
| CFG$_{\text{C-TAG}}$ | 0.10 | 50 | +0.0298 [+0.0077, +0.0516] | 50.9 | +0.0149 [+0.0088, +0.0212] | 51.3 |
| CFG$_{\text{C-TAG}}$ | 0.15 | 50 | +0.0484 [+0.0239, +0.0727] | **51.3** | +0.0205 [+0.0139, +0.0278] | **51.4** |

While the main paper (Table 4) focuses on validating the fundamental properties of TAG under a controlled and conservative guidance regime (e.g., low classifier-free guidance scales $\omega$), it is also important to verify its *practical utility* in high-fidelity settings commonly used in text-to-image generation. To this end, we extend our evaluation to a standard practical setup: Stable Diffusion v2.1 (Rombach et al., 2022) with DDIM (Song et al., 2020a) at a fixed `cfg_scale` of 7.5.

In this high-quality regime, a single metric is often insufficient to capture subtle trade-offs among low-level appearance, geometry, and semantic faithfulness. Therefore, we report a broader metric suite—FID (Heusel et al., 2017), CLIPScore (Hessel et al., 2021), FD-DINOv2 (Stein et al., 2023), and ImageReward (Xu et al., 2023)—to characterize how TAG affects both distribution-level similarity and preference/alignment behavior in this optimized setting (Table 10). This section discusses how these metric families can exhibit different trends under prompt-constrained sampling, even though TAG is applied without additional models or UNet evaluations.

### C.1 VISUAL INTUITION: TEXTURAL FIDELITY VS. GEOMETRIC COHERENCE

Figure 11 provides qualitative intuition for why different metric families may not improve simultaneously in conditional generation. In this example, TAG (left) yields a geometrically more coherent and human-plausible configuration, notably in the object placement relative to the window frame. In contrast, the baseline (middle) exhibits stronger low-level appearance cues (e.g., glass/reflection textures) but shows a mild geometric/occlusion inconsistency, such as an implausible overlap with the frame.

This comparison suggests a difference in metric sensitivity: preference-/alignment-oriented metrics (CLIPScore, ImageReward) typically reward prompt-consistent, human-preferred configurations, whereas distribution-level metrics (FID, FD-DINOv2), which summarize perceptual feature statistics over the entire sample set, may respond more strongly to low-level appearance similarity than to local geometric plausibility.

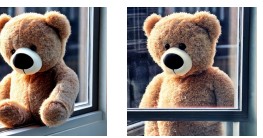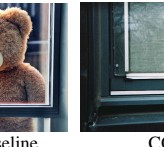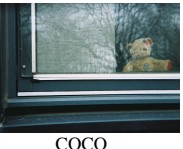

TAG    Baseline    COCO

Figure 11: **Qualitative illustration of a metric trade-off.** Compared to the baseline, TAG produces a more human-plausible scene geometry with consistent object placement relative to the window frame, although it may deviate from the reference in low-level appearance cues (e.g., glass/reflection textures). This example motivates reporting both alignment/preference metrics (ImageReward, CLIPScore) and distribution-level feature metrics (FID, FD-DINOv2), which can respond differently to changes in geometric coherence versus texture fidelity under set-level evaluation.

## C.2 QUANTITATIVE ANALYSIS

**Alignment–distribution trade-off.** Table 10 shows that TAG yields consistent gains on alignment/preference-oriented metrics (CLIP-Score and ImageReward), while FID and FD-DINOv2 mildly increase as $\eta$ grows. We interpret this as a *small trade-off*. In the high-CFG regime, the baseline sampler already produces samples whose feature statistics are relatively *close to those of the reference set* in aggregate. In this context, TAG biases the update toward stricter prompt-consistent behavior, improving *per-prompt faithfulness and preference*, at the cost of a slight shift in dataset-level feature distributions. Consequently, distribution-level

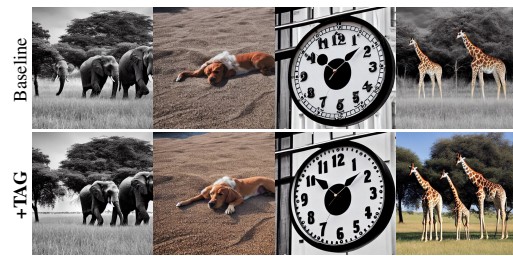

**Conditional Gen.** with SD2.1 ($\omega = 7.5$)

Figure 12: **Qualitative comparison of TAG for conditional generation.** TAG enhances semantic detail and coherence of samples from SD2.1 (Rombach et al., 2022).

metrics should be viewed as aggregate summaries, and are most informative when reported alongside alignment and preference measures.

**Paired evaluation and robustness.** To control for prompt difficulty and sample mismatch, we additionally conduct a *paired* evaluation on the intersection set of prompts, reported in Table 11 with bootstrap 95% confidence intervals (CI) and win-rates.

- **ImageReward:** TAG shows a positive mean gain for all $\eta$. The best setting ($\eta = 0.15$) achieves $\Delta\text{ImageReward} = +0.0205$ (95% CI $[+0.0139, +0.0278]$) with a win-rate of 51.4%.
- **CLIPScore:** A directional improvement is also observed at $\eta = 0.15$ with $\Delta\text{CLIPScore} = +0.0484$ (95% CI $[+0.0239, +0.0727]$).

These paired results support that TAG provides consistent gains in prompt adherence and human preference relative to the same baseline sampling pipeline.

## C.3 SUMMARY

Overall, these extended experiments support the practical utility of TAG in high-fidelity regimes. Across metrics, TAG improves alignment and preference measures (ImageReward and CLIPScore) without additional inference cost, yielding a favorable trade-off: it provides consistent gains in per-instance faithfulness to the conditioning signal while maintaining a competitive distributional profile relative to the baseline.

## D    COMPUTATIONAL COMPLEXITY ANALYSIS

To strictly evaluate the efficiency of the proposed method, we quantify the computational overhead relative to standard sampling and Perturbed Attention Guidance (PAG) (Ahn et al., 2024a). We utilize the `DeepSpeed Flops Profiler` (Rajbhandari et al., 2020) to measure theoretical operations and benchmark real-world inference speeds on an NVIDIA RTX 4090 GPU.

### D.1    EMPIRICAL OVERHEAD

The computational cost of diffusion sampling is dominated by the UNet backbone. As shown in Table 12, the standard baseline (SD v1.5) contains approximately 0.86B parameters, consuming 2.61 GB of VRAM with an inference latency of 0.919 seconds per image.

**PAG (Perturbed Attention Guidance)** requires constructing a perturbed attention path to form the negative guidance signal. This necessitates a second forward pass through the UNet at every denoising step (or doubling the batch size). Consequently, as confirmed in our benchmarks, PAG nearly doubles the memory footprint (2.61 GB $\rightarrow$ 4.95 GB) and increases latency by approximately 112% (0.92s $\rightarrow$ 1.96s).

**TAG (Ours)**, in contrast, operates directly on the latent vectors produced by a single UNet pass. It does not require an auxiliary forward pass. The projection and decomposition operations involve only element-wise linear algebra ($\mathcal{O}(N)$), which is computationally negligible compared to the quadratic complexity of the attention layers in the backbone.

Empirically, TAG maintains the same peak memory usage as the baseline (2.61 GB) and incurs only a minor latency increase ($\sim$ 0.08s total), attributed to the vector projection overhead. This makes TAG significantly more efficient than dual-path guidance methods like PAG.

Table 12: **Computational Cost Analysis.** Measurements were conducted on an NVIDIA RTX 4090 using the DeepSpeed Profiler. FLOPs represents the estimated floating-point operations per image.

| Method | Params | FLOPs | VRAM | Latency | Overhead |
|---|---|---|---|---|---|
| PAG (Ahn et al., 2024a) | 0.86 B | $\approx$ 82 T | 4.95 GB | 1.956 s | **+112.8%** |
| **TAG (Ours)** | 0.86 B | $\approx$ 41 T | 2.61 GB | 1.005 s | +9.4% |
| Baseline | 0.86 B | 40.99 T | 2.61 GB | 0.919 s | – |

# E IMPLEMENTATION OF THE TANGENTIAL AMPLIFYING GUIDANCE

**Algorithm 3** Code: Tangential Amplifying Guidance (TAG)

```
output = scheduler.step(noise_pred, t, latents, return_dict=False)

if apply_tag:
    post = latents
    eta_v, eta_n = t_guidance_scale, 1

    v_t = post / (post.norm(p=2, dim=(1,2,3), keepdim=True) + 1e-8)

    latents = output
    delta = latents - post
    a     = (delta * v_t).sum(dim=(1,2,3), keepdim=True)

    u_n = a * v_t
    u_t = delta - u_n
    latents = post + eta_v * u_t + eta_n * u_n
else:
    latents = output
```

**Algorithm 4** Code: Conditional Tangential Amplifying Guidance (C-TAG)

```
def proj_par(z, n):
    return (z * n).sum(dim=(1,2,3), keepdim=True) * n

def proj(z, v):
    v = v / (v.norm(p=2, dim=(1,2,3), keepdim=True) + 1e-8)
    return (z * v).sum(dim=(1,2,3), keepdim=True) * v

eps_u, eps_c = HeadToEps(noise_pred, latents, t, scheduler, do_cfg)

s_u = -eps_u / (sigma + 1e-12)
s_c = -eps_c / (sigma + 1e-12)

n = latents / (latents.norm(p=2, dim=(1,2,3), keepdim=True) + 1e-8)

g       = s_c - s_u
t_c     = s_c - proj_par(s_c, n)
t_u     = s_u - proj_par(s_u, n)
g_aligned = proj(s_c, t_c - t_u)
g       = g + t_guidance_scale * g_aligned

s_star  = s_u + guidance_scale * g
eps     = -sigma * s_star

model_out = EpsToHead(eps, latents, t, scheduler)
latents = scheduler.step(model_out, t, latents, return_dict=False)
```

## F ADDITIONAL QUALITATIVE RESULTS

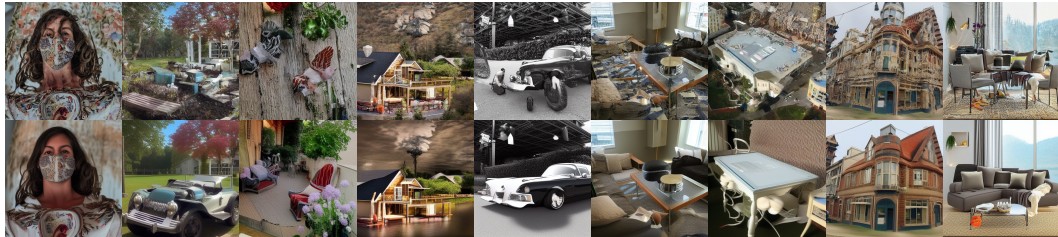

*Stable Diffusion 1.5*

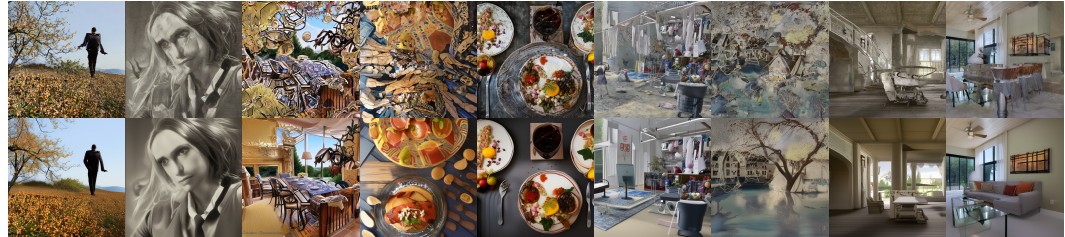

*Stable Diffusion 2.1*

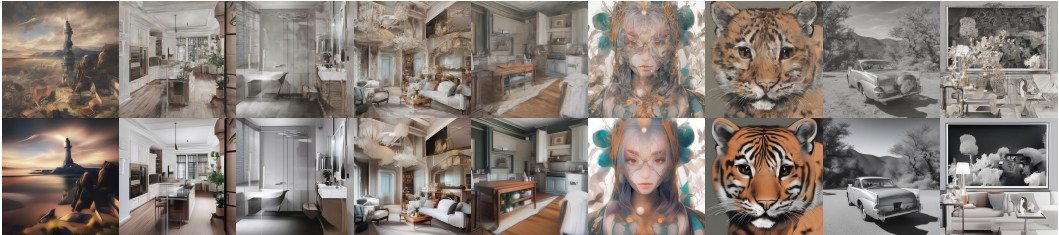

*Stable Diffusion XL*

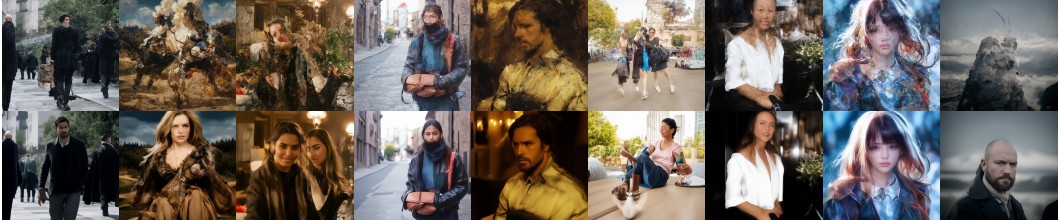

*Stable Diffusion 3*

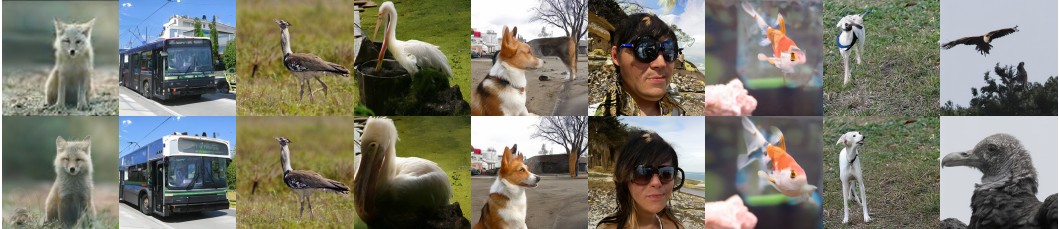

*EDM2*

Figure 13: **Qualitative results for unconditional generation across backbones.** For each model (SD1.5/2.1 (Rombach et al., 2022), SDXL (Podell et al., 2024), SD3 (Esser et al., 2024) and EDM (Karras et al., 2024b)), the top row shows baseline sampling and the bottom row shows +TAG at matched NFEs. TAG yields sharper, more coherent structure with fewer artifacts while preserving diversity.

## F.1 CONDITIONAL GENERATION WITH SDXL

(a) Cups, fruit and a person in a white t shirt

(b) An interesting fruit tree bare of leaves stands in front of a lake near an apartment building.

(c) an image of a woman that is next to a boy

(d) a little kid all dressed up in front of construction equipment

Figure 14: **Qualitative examples 1/4 of CFG-based T2I generation** on the MSCOCO 2014 validation set using SDXL (Podell et al., 2024), APG (Sadat et al., 2025), TCFG (Kwon et al., 2025) and our C-TAG method.

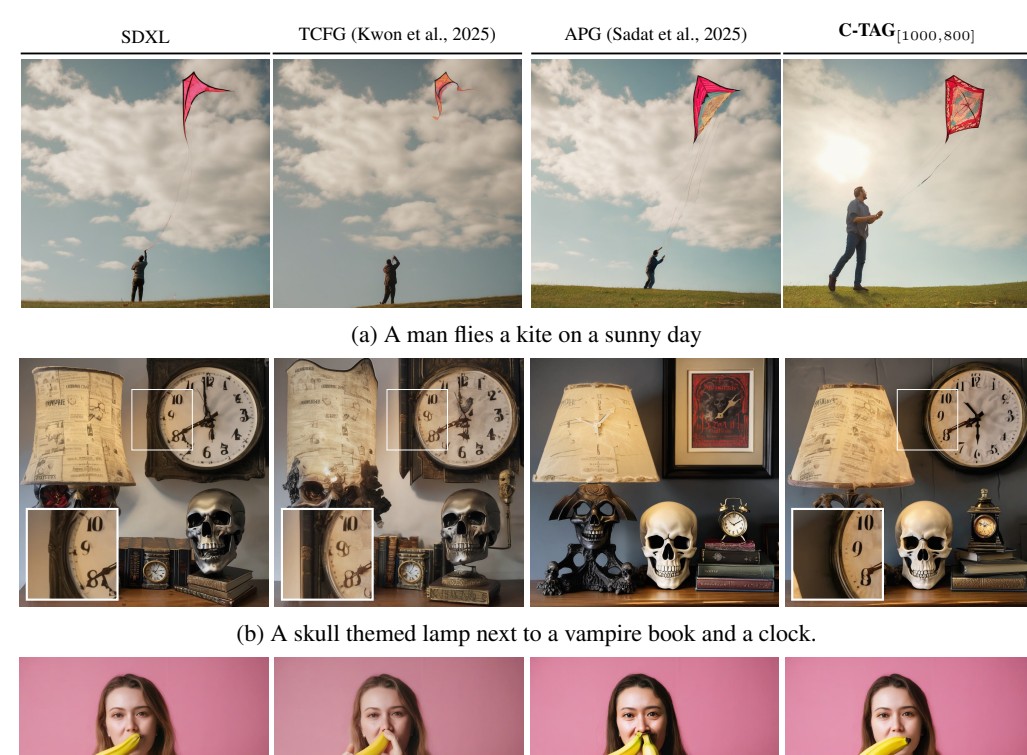

| SDXL | TCFG (Kwon et al., 2025) | APG (Sadat et al., 2025) | **C-TAG**$_{[1000,800]}$ |

(a) A man flies a kite on a sunny day

(b) A skull themed lamp next to a vampire book and a clock.

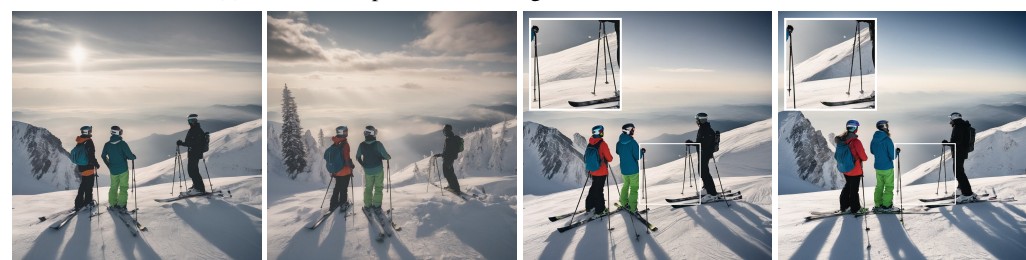

(c) A woman in pink shirt holding a banana in front of her face.

(d) Skiers admiring the view from the top of the mountain

Figure 15: **Qualitative examples 2/4 of CFG-based T2I generation** on the MSCOCO 2014 validation set using SDXL (Podell et al., 2024), APG (Sadat et al., 2025), TCFG (Kwon et al., 2025) and our C-TAG method.

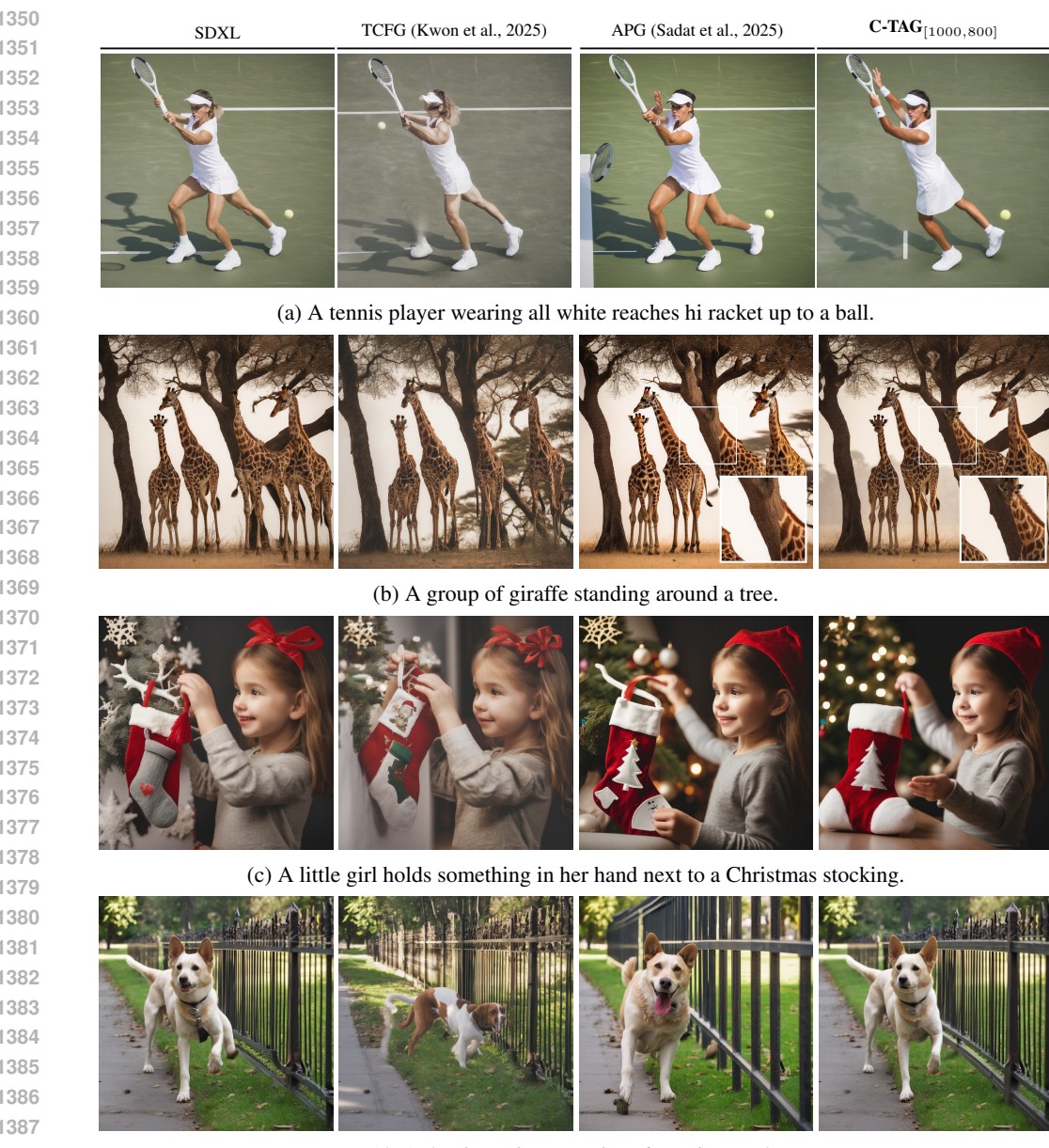

Figure 16: **Qualitative examples 3/4 of CFG-based T2I generation** on the MSCOCO 2014 validation set using SDXL (Podell et al., 2024), APG (Sadat et al., 2025), TCFG (Kwon et al., 2025) and our C-TAG method.

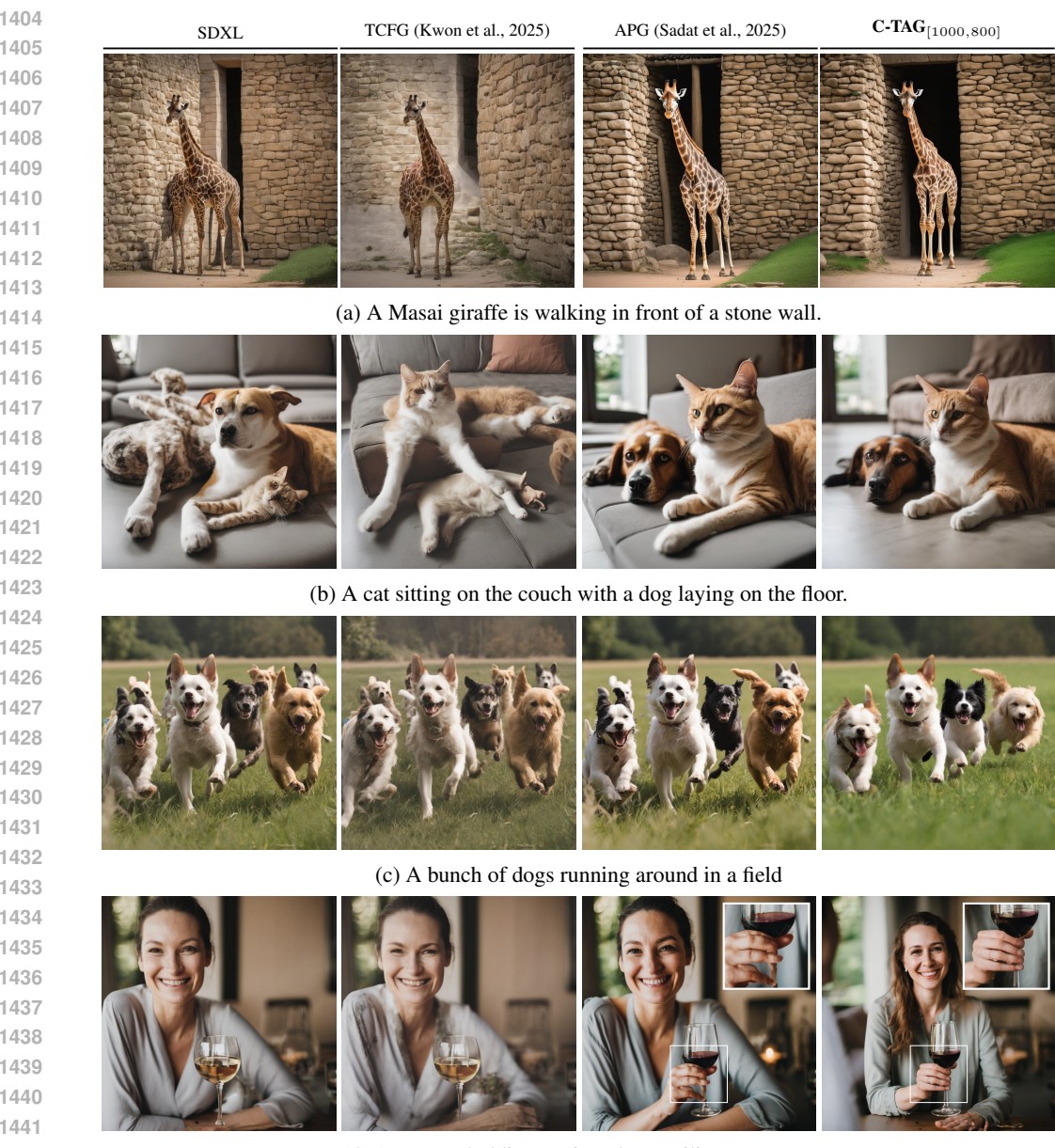

SDXL     TCFG (Kwon et al., 2025)     APG (Sadat et al., 2025)     **C-TAG**$_{[1000,800]}$

(a) A Masai giraffe is walking in front of a stone wall.

(b) A cat sitting on the couch with a dog laying on the floor.

(c) A bunch of dogs running around in a field

(d) A woman holding a wine glass smiling at camera

Figure 17: **Qualitative examples 4/4 of CFG-based T2I generation** on the MSCOCO 2014 validation set using SDXL (Podell et al., 2024), APG (Sadat et al., 2025), TCFG (Kwon et al., 2025) and our C-TAG method.

## G    LLM USAGE DISCLOSURE

During the preparation of this paper, the authors made limited use of large language models (LLMs) for polishing the writing, grammar refinement and LaTeX formatting. LLMs were not used for generating research ideas, designing or conducting experiments, analyzing results, or formulating conclusions. All scientific content and contributions are entirely the responsibility of the authors, and any LLM-assisted text was carefully reviewed and revised before inclusion.

## H    REPRODUCIBILITY STATEMENT

We implemented and evaluated our models and all baselines using PyTorch (Paszke et al., 2019), HuggingFace Diffusers (von Platen et al., 2022), and the official implementations and documentation for EDM2 (Karras et al., 2024b) and SV3D (Voleti et al., 2024).

