# OpenReview forum: "TAG: Tangential Amplifying Guidance for Hallucination-Resistant Diffusion Sampling"
_ICLR.cc/2026/Conference — Submitted to ICLR 2026_

### Official Review · Reviewer_RZdQ · 2025-10-25

**Soundness:** 3
**Presentation:** 4
**Contribution:** 3
**Rating:** 8
**Confidence:** 4

**Summary:**

This paper introduces Tangential Amplifying Guidance (TAG), a training-free, plug-and-play inference method for diffusion samplers. At each solver step, the base update is decomposed into a normal (radial) component—aligned with the current latent and a tangential (manifold-following) component. TAG keeps the normal part unchanged (to respect the scheduler’s radius/SNR trajectory) and amplifies the tangential part to steer trajectories along higher-density regions and reduce hallucinations. The paper is motivated from Tweedie’s formula and formalizes it via a first-order Taylor analysis, proving that increasing the tangential weight monotonically increases a local log-likelihood gain for deterministic samplers (Theorem 4.1). Practically, TAG claims to consistently improve unconditional and conditional generation on various models (SD-1.5/2.1/XL/3), with no extra NFEs and tiny runtime deltas. Also, the method composes with prior inference-time guidance (SAG, PAG, SEG).

**Strengths:**

1. The paper is well-written and diagramed properly.
2. The experiments and empirical observations are coherent with proposed theroem and methods. Also the paper raises well-stated limitations, that too-large η or normalization can harm quality by disturbing radius calibration.
3. The proposed method has broad applicability, composes with existing guidance on multiple samplers (DDIM, DPM++) and with flow‑matching model (SD3) with noticeable quality improvement

**Weaknesses:**

1. Although the paper does cited geometry‑aware guidance like APG (Sadat, 2024) and TCFG (Kwon, 2025). A direct and controlled comparison (same models/samplers/NFEs) is currently absent.
2. More visual examples are needed to show how the generation result changes after applying C-TAG (Especially on CFG, which is the most commonly used in industry). In the final version, please include a batch of examples comparing the sampling with CFG and C-TAG to better demonstrate the improvement on image quality and diversity.

**Questions:**

1. Does emphasizing high probability regions via TAG reduce the diversity of generated samples ?
2. Another 2025 paper uses “Temporal Alignment Guidance (TAG)” for a different on-manifold mechanism; recommend a distinct name to avoid confusion
3. The paper claims negligible overhead, can the authors quantify the overhead relative to standard sampling and compare it to other guidance methods?

---

> ### Author Response · Authors · 2025-11-21
>
> **Dear reviewer RZdQ,**
>
> We sincerely thank you for your very positive evaluation and for the detailed, constructive suggestions. We have revised the manuscript accordingly, and we respond to each point below with references to the updated sections.
>
> ---
>
> ### **Comment 1**
>
> > **A direct and controlled comparison with geometry-aware guidance methods (e.g., APG, TCFG) under identical settings is missing.**
>
> **Response 1**
>
> Thank you for highlighting this. In the revised manuscript, we added **fully controlled comparisons with APG and TCFG under matched settings** (same SDXL backbone, sampler, NFEs, CFG scale, and evaluation protocol), reported in **Table 5 (Page 9)**.
>
> The key takeaway is that, under a direct head-to-head evaluation with matched settings, C-TAG achieves performance comparable to APG/TCFG. Moreover, applying the **fixed windowing schedule** motivated in our limitation analysis (Page 10) yields a improvement in the standalone C-TAG results, leading to stronger performance in this regime. We updated the revised manuscript to make this comparison explicit, and we include side-by-side qualitative results (Appendix F.1, Page 24-27) to facilitate direct inspection under matched settings.
>
> ---
>
> ### **Comment 2**
>
> > **Please include more visual examples comparing CFG vs C-TAG to better illustrate the qualitative effect.**
>
> **Response 2**
>
> We agree and have substantially expanded the qualitative evidence. The revised paper now includes **side-by-side CFG vs CFG+C-TAG examples using SDXL in the main text (Figure 6, Table 5)**, and **larger batches in the appendix (Appendix F.1, Page 24-27)**.
>
> Across diverse prompts, TAG **reduces hallucinations and spurious artifacts** (e.g., unrealistic objects or attributes not implied by the prompt) while **improving fidelity and semantic coherence**. The expanded grids are intended to make this effect clear in an industry-standard CFG setting.
>
> ---
>
> ### **Comment 3 (Questions)**
>
> > **(Q1) Does TAG reduce diversity?
> > (Q2) Another paper uses “TAG” for a different guidance; please avoid confusion.
> > (Q3) Please quantify overhead relative to standard sampling and other guidance methods.**
>
> **Response 3**
>
> **(Q1) Diversity.** Empirically, we observe that TAG improves FID while maintaining or increasing Inception Score across our benchmarks (Tables 1–3, Page 6). These metrics suggest that the diversity of the generated distribution is largely preserved, as mode collapse typically correlates with a significant decrease in IS. Qualitatively, the expanded samples in Figures 14-17 (Page 24-27) indicate that TAG retains variation across prompts while primarily correcting implausible artifacts. We believe that TAG acts essentially as a mechanism for hallucination suppression rather than diversity suppression: because it rescales the tangential component of the existing solver update without injecting rigid external constraints, it aims to refine the trajectory toward the manifold without stifling the intrinsic variation of the sampling process.
>
> **(Q2) Naming.** Thank you for noting the acronym overlap. We are considering a **manifold-forward** naming.
>
> **(Q3) Overhead.** We added a detailed overhead analysis in **Appendix D (Page 21)**, including FLOPs/VRAM/latency benchmarks versus baseline and PAG [A1]. TAG requires **no extra UNet evaluations** and only inexpensive vector projections, resulting in **minor overhead** (single-digit %) compared to baseline and **dramatically less overhead than dual-path guidance such as [A1]**.
>
> ### **References**
> > [A1] Ahn, Donghoon, et al. "Self-rectifying diffusion sampling with perturbed-attention guidance." European Conference on Computer Vision. Cham: Springer Nature Switzerland, 2024.
>
> ---
>
> We sincerely thank you again for the insightful feedback. Your suggestions directly motivated the controlled APG/TCFG comparisons, expanded CFG-focused qualitative results, and clearer overhead reporting, all of which strengthened the revised manuscript.

---

> > ### Author Response · Authors · 2025-11-28
> > **Kind reminder (Paper #14964)**
> >
> > Dear Reviewer RZdQ,
> >
> > Thank you again for your time and effort in reviewing our manuscript. We have posted our response addressing your concerns and suggestions.
> >
> > If you have any additional questions or require further clarification, we are happy to discuss them. We eagerly await your valuable feedback.
> >
> > Best regards,
> >
> > Authors of Submission #14964

---

### Official Review · Reviewer_en3y · 2025-10-29

**Soundness:** 2
**Presentation:** 2
**Contribution:** 1
**Rating:** 0
**Confidence:** 3

**Summary:**

The authors propose *TAG* (Tangential Amplifying Guidance) a technique which aims to improve guidance mechanisms in diffusion models and steer the guided samples towards high probability regions. This is accomplished by splitting the guidance term into orthogonal and tangential components and applying a guidance factor $\eta \geq 1$ to the tangential component. The authors then perform some experiments with unconditional and conditional generation.

**Strengths:**

* I like that the authors also include experiments on flow matching models.
* The author put a lot of details into explaining their algorithm.
* I like that the authors report the CLIP-score.
* The quality of writing is generally good.

**Weaknesses:**

## Primary concerns
I highlight my concerns with the manuscript below.

### Novelty
I have several concerns regarding TAG and it's novelty within the literature.

* Although the authors mention [1, 2] in the last paragraph of Section 2 they don't really compare to them experimentally or in text. Both of these works seem very similar to the proposed TAG method. In particular the update equation in (6) seems very similar to the update equation $\Delta D_t(\eta) = \Delta D_t^\perp + \eta \Delta D_t^{\|}$ in [1, Section 4].
* The work of [3] also seems highly relevant as they also breakdown into guidance via orthogonal projections.
* Likewise with [4].
* The idea of TAG also seems to be related to works exploring manifold guidance [5] and in particular [6].
* Lots of papers working on inverse problems seem to explore similar ideas, e.g., [5,8] to name a few.
* Authors should also mention the analysis of [9].

### Implementation and theory
* Shouldn't the normal component be labelled $\boldsymbol P_k^\perp$? The projection $\boldsymbol P_k = \mathbf x_k \mathbf x_k^\top$ should be the projection onto the tangent space of $\mathbf x_k$ and not the normal component. Thus I have concerns about the rest of the paper which modifies the the "tangential" component $\boldsymbol P_k^\perp$ via $\eta$. Isn't this just modifying the normal component?
* Further analysis in the paper seems to indicate that $\boldsymbol P_k^\perp$ is the normal projection *a la* line 244. So now I have **deep concerns** about the soundness of the proposed method.


### Empirical
I do not find the empirical results in Section 5 to be satisfactory and I believe there are numerous shortcomings which I outline below.

* My primary issue is the lack of comparison to other techniques which look at orthogonal projection like [1-2].
* I also have concerns that the baselines are unusually poor. For example Figures 5 and 6. Despite, their shortcomings these models can work with CFG or unconditional guidance and produced images for the baselines seem exceptionally poor.
* I am not sure why the authors study unguided generation when TAG is meant for *guidance*.
* While I appreciate the breadth of experiments I feel that comparison to more *relevant* prior works is required.

## Minor concerns or comments
* I dislike the notation
$$\begin{equation}
d \mathbf x_k = \mathbf f(\mathbf x_k, t) dt_k + g(t_k) d \mathbf W_{t_k}
\end{equation}$$
because the equation describes an Ito SDE and not the discretized numerical scheme for which you would have timesteps $\{t_k\}$.
* It would be good to also include the Image Reward metric which has become popular in evaluating conditional generation [7].


## References
[1] Sadat, Seyedmorteza, Otmar Hilliges, and Romann M. Weber. "Eliminating oversaturation and artifacts of high guidance scales in diffusion models." The Thirteenth International Conference on Learning Representations. 2024. https://arxiv.org/pdf/2410.02416

[2] Kwon, Mingi, et al. "TCFG: Tangential Damping Classifier-free Guidance." Proceedings of the Computer Vision and Pattern Recognition Conference. 2025. https://openaccess.thecvf.com/content/CVPR2025/papers/Kwon_TCFG_Tangential_Damping_Classifier-free_Guidance_CVPR_2025_paper.pdf

[3] Zheng, Candi, and Yuan Lan. "Characteristic guidance: Non-linear correction for diffusion model at large guidance scale." ICML 2024. https://arxiv.org/pdf/2312.07586v5

[4] Armandpour, Mohammadreza, et al. "Re-imagine the negative prompt algorithm: Transform 2d diffusion into 3d, alleviate janus problem and beyond." arXiv preprint arXiv:2304.04968 (2023). https://arxiv.org/pdf/2304.04968

[5] He, Yutong, et al. "Manifold Preserving Guided Diffusion." The Twelfth International Conference on Learning Representations. https://arxiv.org/pdf/2311.16424

[6] Sun, Shikun, et al. "SDDM: score-decomposed diffusion models on manifolds for unpaired image-to-image translation." International Conference on Machine Learning. PMLR, 2023. https://proceedings.mlr.press/v202/sun23n/sun23n.pdf

[7] Xu, Jiazheng, et al. "Imagereward: Learning and evaluating human preferences for text-to-image generation." Advances in Neural Information Processing Systems 36 (2023): 15903-15935.

[8] Boys, Benjamin, et al. "Tweedie Moment Projected Diffusions for Inverse Problems." Transactions on Machine Learning Research. https://openreview.net/pdf/6762157974f5480b3bf64dc61ed3b618a6e7772e.pdf

[9] Stanczuk, Jan, et al. "Your diffusion model secretly knows the dimension of the data manifold." arXiv preprint arXiv:2212.12611 (2022). https://arxiv.org/pdf/2212.12611

**Questions:**

1. How do you differentiate from prior works which also explore orthogonal projections for guidance?
2. Does TAG work for higher-order schemes?
3. Shouldn't the normal component be labelled $\boldsymbol P_k^\perp$?

---

> ### Author Response · Authors · 2025-11-21
>
> **Dear reviewer en3y,**
>
> We sincerely appreciate your careful reading of our manuscript and your constructive comments. Below we respond to each point; we address **Comment 2** first in order to clarify a potential misunderstanding about our notation and geometric interpretation.
>
> ---
>
> ### **Comment 2**
>
> > **Implementation & Theory**: Shouldn't the normal component be labelled $\boldsymbol P_{k}^{\perp}$?
>
> **Response 2**
> We agree that our notation can be confusing at first glance. We define
> $$
> \boldsymbol P(\boldsymbol x)=\widehat{\boldsymbol x}\widehat{\boldsymbol x}^{\top},\quad
> \boldsymbol P^{\perp}(\boldsymbol x)=\boldsymbol I-\boldsymbol P(\boldsymbol x),
> $$
> where $\widehat{\boldsymbol x}=\boldsymbol x/|\boldsymbol x|_2$.
> If 'normal/tangent' were defined **with respect to the vector $\boldsymbol x$**, then $\boldsymbol P(\boldsymbol x)$ would naturally correspond to the tangent (along $\boldsymbol x$) and $\boldsymbol P^\perp(\boldsymbol x)$ to the normal (orthogonal to $\boldsymbol x$).
>
> In our paper, however, 'normal/tangential are defined **with respect to the iso-noise manifold $\mathcal M_k$** at $\boldsymbol x_k$: the tangential component is the one that moves *along* $\mathcal M_k$. With this convention, $\boldsymbol P_k^\perp$ corresponds to the tangential direction in the regime where $\mathcal M_k$ is well-approximated.
>
> **E1. High-noise regime (Gaussian annulus).**
> For large $t_k$, $p_k$ is close to an isotropic Gaussian $N(0,\sigma_k^2 I)$. By the Gaussian annulus theorem, most mass concentrates near the shell $|\boldsymbol x_k|\approx \sigma_k \sqrt d$, so $\mathcal M_k$ is well approximated by a sphere. On a sphere, the radial direction $\widehat{\boldsymbol x}_k$ is the unit normal and its orthogonal complement is tangent. Hence
>
> $$
> \boldsymbol P_{\mathcal M_k}^{\perp}(\boldsymbol x_k)
> :=\boldsymbol I-\widehat{\boldsymbol x}_k \widehat{\boldsymbol x}_k^{\top}
> $$
>
> acts as an approximately **tangential projector to $\mathcal M_k$** in the mid/high-noise regime.
>
> **E2. Low-noise regime and why artifacts appear.**
> As $t_k \to 0$, $\mathcal M_k$ inherits the geometry of the data manifold $\mathcal M_0$. Under standard assumptions, [A1] shows that the score aligns with the **normal bundle of $\mathcal M_0$**, so the true tangential directions become orthogonal to the score rather than to $\widehat{\boldsymbol x}\_k$. Therefore $\boldsymbol P_k^\perp(\boldsymbol x_k)$ can misalign with $\mathcal T_{\boldsymbol x_k}\mathcal M_k$ at low noise, which explains the artifacts in Figure 2.
>
> To mitigate this, we added a **time-dependent windowed TAG** that turns off tangential amplification in the low-noise regime. Using a window $w(t_k)\in[0,1]$, we set $\tilde\eta_k=w(t_k)\eta_k$ so that $w(t_k)\approx 1$ for mid/high noise and $w(t_k)\to 0$ as $t_k \to 0$. Empirically, this windowed variant improves performance (Table 8).
>
> ### **Revisions in the manuscript.**
>
> 1. **Notation clarified:** we now write $\boldsymbol P_{\mathcal M_k}^{\perp}(\boldsymbol x_k)$ to make the manifold-relative meaning explicit.
> 2. **Geometry discussion expanded:** we added a concise explanation linking the Gaussian-annulus (high-noise) intuition with the low-noise score-normal behavior in [A1], to clarify when $\boldsymbol P_k^\perp$ is a faithful tangential projector. (Section 3, 6)
>
> ### **References**
> > [A1] Stanczuk, Jan, et al. "Your diffusion model secretly knows the dimension of the data manifold." arXiv preprint arXiv:2212.12611 (2022).
>
> ---

---

> ### Author Response · Authors · 2025-11-21
>
> ### **Comment 1 (Part 1: Clarification of Related Works):**
>
> > **Novelty**: How do you differentiate from prior works which also explore orthogonal projections for guidance?
>
> **Response 1**
>
> Thank you for the question. Below we briefly summarize how representative prior works define and use “tangential/orthogonal” directions for guidance, and then clarify how TAG differs.
>
> ### **[1. Eliminating oversaturation in CFG]**
> **Problem**
>
> * High CFG can cause oversaturation and artifacts.
>
> **Observation**
>
> * The CFG update component **parallel to the conditional score** is the main driver of saturation.
>
> **Solution**
>
> * **Dampen the parallel component** of the CFG update (keep the orthogonal part), via a weighted split with $\eta \le 1$.
>
> ### **[2. Tangential Damping CFG]**
> **Problem**
>
> * Misalignment between conditional and unconditional scores yields off-manifold samples.
>
> **Observation**
>
> * The misalignment is concentrated in **low-SVD (tangential) directions**.
>
> **Solution**
>
> * **Project the unconditional score** onto the shared high-SVD subspace (approx. normal to $\mathcal M_t$), effectively suppressing the low-SVD/tangential part.
>
> ### **[3. Characteristic Guidance]**
> **Problem**
>
> * High CFG produces oversaturation / artifacts (same symptom as [1]).
>
> **Observation**
>
> * CFG ignores a nonlinear correction term in the Fokker–Planck dynamics, causing misaligned updates.
>
> **Solution**
>
> * **Approximate the corrected guided score** via characteristics + fixed-point iteration; any projection operator is only a computational regularizer (ideally identity), not the core mechanism.
>
> ### **[4. Perp-Neg]**
> **Problem**
>
> * Negative prompts can cancel desirable features when overlapping with positive prompts.
>
> **Observation**
>
> * The negative score contains components aligned with the positive score that inadvertently remove target attributes.
>
> **Solution**
>
> * Apply negative guidance only along the direction orthogonal to the positive score, so shared positive features are not suppressed.
>
> ### **[5. Manifold-Preserving Guided Diffusion (MPGD)]**
> **Problem**
>
> * Loss-based guidance (e.g., DPS-style) can push samples off-manifold.
>
> **Observation**
>
> * Applying guidance directly on $x_t$ may violate manifold constraints.
>
> **Solution**
>
> * Apply guidance on the clean estimate $x_{0|t}$ and **restrict updates to $\mathcal T_{x_{0|t}}\mathcal M_0$** (using an external AE to model $\mathcal M_0$).
>
> ### **[6. Score-Decomposed Diffusion Models on Manifolds (SDDM)]**
> **Problem**
>
> * Naively applying I2I guidance to the score entangles refinement and transport, hurting quality.
>
> **Observation**
>
> * Generally, decomposing the score into tangential vs normal components is **typically considered ill-defined or meaningless**.
> * In contrast to general properties, under the **conditional I2I setting** where a reference image $y_{0}$ induces compact and well-separated reference-dependent manifolds (i.e., $\mathcal M_t(y_0)$), SDDM decomposes the score as $s(x,t) = s_{r}(x,t) + s_{d}(x,t)$ with respect to $\mathcal M_{t}$ and **treats the tangential component $s_{r}(x,t)$ as a refinement part on the manifold**, while the normal component participates in mapping between manifolds of adjacent time steps
>
> **Solution**
>
> * **Project score and I2I guidance onto $\mathcal T_{x_t}\mathcal M_t$** for on-manifold refinement, while the normal component handles transitions between $\mathcal M_t$’s.
>
> ---
>
> To more thoroughly recognize the relevant literature and clarify TAG’s contributions, we have included a detailed discussion in **Appendix. E (page 21, 22)** and updated the **Related Work** section to explicitly present the geometry-related preliminaries (page 2, beginning around **line~104**).

---

> > ### Author Response · Authors · 2025-11-21
> >
> > ### **Comment 1 (Part 2: Our Contributions):**
> >
> > Now we summarize our contribution and clarify how TAG differs from the above works.
> >
> > ## **TAG: Tangential Amplifying Guidance**
> >
> > **Problem.**
> > Diffusion models still generate hallucinations and off-manifold artifacts. Many inference-time guidances improve fidelity, but they are typically either
> > (i) **tightly tied to CFG-style conditional guidance** and its specific score algebra (e.g., [1–4]), or
> > (ii) **computationally heavier**, requiring extra UNet evaluations or dual-path passes.
> > Meanwhile, geometry-aware approaches that explicitly leverage diffusion intrinsic geometry are often **task- or condition-specific**—e.g., inverse problems based on DPS-style observations [5]—or limited to narrow settings such as unpaired I2I with reference-dependent manifolds [6].
> >
> > **Our key premise** is that diffusion sampling admits an **intrinsic geometric structure** that can be exploited **generally**, including in unconditional and standard conditional generation, without relying on CFG algebra or extra network passes.
> >
> > ### **Motivation.**
> >
> > Starting from Tweedie’s formula, we consider the single-step solver update $\Delta_k$ as a direction that should move samples toward higher-probability regions. To isolate the geometry induced by $\boldsymbol x_k$, we define
> > $\boldsymbol P_k$ and $\boldsymbol P_k^\perp$, which correspond to the **radial (normal)** and **tangential** projections **with respect to the iso-noise manifold $\mathcal M_k$** at $\boldsymbol x_k$.
> >
> > ### **Observation.**
> >
> > Empirically, the **tangential projection** $\boldsymbol P_k^\perp \Delta_k$ consistently captures rich semantic structure, whereas the full update $\Delta_k$ contains substantial unstructured artifacts (Figure 2).
> >
> > ### **Solution.**
> >
> > We therefore amplify only the tangential component at each step:
> >
> > $$
> > \Delta_k^{\mathrm{TAG}}
> > = \boldsymbol P_k\Delta_k + \eta\boldsymbol P_k^\perp\Delta_k,
> > \quad \eta \ge 1,
> > $$
> >
> > which strengthens semantic refinement while suppressing off-manifold drift.
> >
> > ## **Main contributions.**
> >
> > 1. **Meaningful tangential/normal split in general diffusion.**
> >    We show that decomposing the **single-step solver update** (equivalently, the score; Eq. 13) into radial and tangential parts yields a **stable, meaningful separation**: the tangential part carries semantic refinement (Fig. 2), while the radial part is largely noisy. This contrasts with prior observations that generic tangential/normal decompositions are **often “generally meaningless”** outside special settings such as I2I reference-induced manifolds [6].
> > 2. **Geometry–likelihood link.**
> >    We connect the score to sampling geometry and prove that tangential amplification increases local log-likelihood via a first-order Taylor argument.
> > 3. **A lightweight, backbone-agnostic method.**
> >    TAG is a training-free, sampler-level plug-in that introduces no extra UNet passes and applies across unconditional, conditional, and modern backbones.
> >
> > ## **Clarification vs. the closest related works.**
> >
> > * **[2] Tangential Damping CFG:**
> >   Projects the **unconditional score** onto a high-SVD subspace derived from cond/uncond scores, suppressing low-SVD (tangential) directions **specifically within CFG**. It does not analyze the semantic role of tangential components along the solver trajectory, nor propose modifying the base solver update itself.
> > * **[5] MPGD / DPS-style methods:**
> >   Preserve manifolds by applying guidance on $x_{0|t}$ and explicitly modeling $\mathcal M_0$ via external autoencoders. These methods target **observation-conditioned inverse problems**, not generic unconditional/standard conditional sampling.
> > * **[6] SDDM:**
> >   Decomposes the score on **reference-dependent manifolds $\mathcal M_t(y_0)$** for unpaired I2I, and explicitly states that tangential/normal splits are **generally meaningless** outside that special conditional structure. TAG’s core contribution is to make such a split meaningful **without** relying on reference-conditioned manifolds.
> >
> > ---
> >
> > We carefully reviewed the works you raised. In relation to each, we believe TAG provides a distinct contribution: it identifies a generally meaningful tangential refinement direction intrinsic to diffusion sampling, proves its likelihood benefit, and turns it into an efficient plug-in guidance. We hope this clarification helps address the novelty concern.

---

> > > ### Author Response · Authors · 2025-11-21
> > >
> > > ### **Comment 3:**
> > >
> > > > **C1. no comparison to other orthogonal-projection techniques like.**
> > > > **C2. Baselines seem unusually poor (e.g., Figs. 5–6).**
> > > > **C3. Unclear why you study unguided generation if TAG is for guidance.**
> > >
> > > **Response 3**
> > >
> > > **E1. Comparison to orthogonal-projection guidance.**
> > > We agree this comparison is essential. In the revised manuscript we added **fully controlled experiments with APG and TCFG under matched settings**. Results are reported in **Table 5** (Page 9).
> > > The key finding is that **TAG improves performance even when attached on top of APG or TCFG**, showing TAG is *complementary* to these projection-based CFG refinements rather than duplicating them.
> > >
> > > **E2. Why some original baselines looked weak, and how we fixed this.**
> > > We understand the concern about the poor-looking samples in Figures 5–6. The high FIDs / visibly weak samples in the original submission mainly came from **unconditional or low-CFG/condition-only settings on SD-family backbones**, which are known to be much weaker than ImageNet-tuned guided models where FID 1–10 is usually reported.
> > > To make the empirical picture literature-faithful, we re-evaluated and added stronger baselines:
> > >
> > > * **SDXL with high-CFG**: baselines are now in the expected FID regime, and TAG still gives consistent gains (Table 5; see also new CFG visual comparisons in Fig. 6 and Appendix Figs. 12–13 (Page 19-20)).
> > > * **SD3 flow-matching**: we corrected the earlier overly weak setup and re-ran unconditional evaluation on **30K ImageNet samples at matched 50 NFEs** (Table 6).
> > > * **EDM2 (modern ImageNet backbone)**: we added TAG on top of **the official EDM2 ImageNet-512 pipeline**, where the baseline FID is already literature-level ($\approx$ 5). TAG continues to improve semantic quality (FD-DINOv2) while preserving competitive FID (Table 7, Page 9).
> > >
> > > **E3. Why we study unguided (unconditional) generation.**
> > > Although TAG is useful as a *guidance* enhancement, it is **not defined through CFG algebra**. TAG operates on the **solver update itself** after a single UNet evaluation, so it applies to:
> > >
> > > * **unconditional sampling**, where it functions as **self-guidance via intrinsic iso-noise geometry**, and
> > > * **conditional sampling**, where it can be layered on top of CFG/PAG/SEG/etc.
> > >
> > > Studying unconditional generation therefore serves two purposes:
> > >
> > > 1. **Isolating the geometric effect** of tangential amplification independent of any external guidance signal; and
> > > 2. **Demonstrating backbone-agnostic applicability**, including to modern architectures such as flow-matching SD3 and EDM2.
> > >
> > > We hope these experiments address your concern.
> > >
> > > ---
> > >
> > > We sincerely thank you again for the thorough and candid feedback. Your comments directly motivated substantial improvements to both our empirical evaluation and presentation. We hope the new results and clarifications address your concerns.

---

> > > > ### Comment · Reviewer_en3y · 2025-11-23
> > > >
> > > > I appreciate the detailed rebuttal by the authors and their efforts to improve their submission. My questions about the notation and lack of clarity stemming from it now, so the contributions now seem to be technically valid. I appreciate the addition of Appendix E which alleviates some of my concerns.
> > > >
> > > > I'm still concerned about the empirical studies and the comparison to prior works. Why is C-TAG appended to other methods in Table 5? Shouldn't they be compared independently?

---

> ### Author Response · Authors · 2025-11-24
>
> **We really appreciate your response.**
>
> ---
>
> ### **Response 1**
> We agree that prior methods should be judged via matched standalone comparisons, and Table 5 is organized to do that. In addition, our method can be viewed as introducing a **sampling strategy**—tangential amplification—that refines the solver’s sampling trajectory by adding a tangential term (controlled by $\eta$) inside each update. Since this operates within the sampling step using the same UNet evaluations, it is **naturally compatible with different guidance choices**.
>
> Under identical settings, Table 5 reports direct head-to-head standalone rows such as
> **$\text{CFG}$ vs. $\text{CFG}_{\mathbf{C\text{-}TAG}}$ vs. $\text{CFG+APG}$ vs. $\text{CFG+TCFG}$**.
> These are independent baselines evaluated under the same backbone, guidance scale, NFEs, and steps.
>
> We additionally include the **$\text{CFG}_{\mathbf{C\text{-}TAG}}+\text{APG/TCFG}$** rows to highlight a practical aspect of the proposed strategy: because C-TAG inserts a lightweight tangential amplification into the sampling update **without changing compute or the underlying guidance formulation**, it can be plugged into existing guidance schemes straightforwardly. Empirically, even when strong geometry-aware guidances such as APG or TCFG are used, adding C-TAG still yields further FID improvements at matched cost, supporting its compatibility across guidance choices.
>
> **[Revision in the paper]**
> Finally, to make the standalone comparison even clearer, new revision adds qualitative side-by-side results in **Appendix D.1–D.2 (Figs. 14–15)**, directly contrasting **$\text{CFG}_{\textbf{C-TAG}}$** against APG and TCFG under matched settings, and also illustrating the effect of composing them.
>
> We hope this clarifies that Table 5 contains the required independent comparisons, while the composed rows are included solely to demonstrate that C-TAG can be naturally used together with existing guidance methods and still improves quality.
>
> ---
>
> ### **Response 2**
> **[Novelty & Notation]**
> We are glad that Appendix E alleviated some of your concerns. If any aspects remain unclear or would benefit from further clarification, we would greatly appreciate your guidance on which specific points to address in the revision.

---

> > ### Comment · Reviewer_en3y · 2025-11-25
> >
> > I reread **Table 5**. I should restate my question from earlier more clearly: why compare $CFG_{C-TAG} + APG$ vs $CFG + APG$ instead of $CFG_{C-TAG}$ vs $CFG + APG$? And *mutatis mutandis* for other prior works.
> >
> > Moreover, I read the concerns raised by **reviewer rNL6** and am considered by the unusually poor baselines, in FID and CLIP Score. Part of this might be alleviated from reporting FD-Dinov2 (which the authors report later) in addition to FID and Image Reward with Clip score, bu the numbers seem rather poor, especially for Clip score in Table 4. Also why isn't Clip score reported in Table 5?
> >
> > I think Tables 1 - 2 could be strengthened by comparing to more compelling baselines in addition to DDIM. Also in Table 3, DPM++ is just the euler scheme for data prediction whereas DDIM is the euler scheme for noise prediction. Investigated the higher-order schemes both DPM++2S or DPM++2M could be interesting along with noise prediction formulations.
> >
> > Currently, I agree with **reviewer rNL6** and will keep my current recommendation and encourage the authors to improve their experimental studies, in particular comparison to prior works and baselines.

---

> > > ### Author Response · Authors · 2025-11-28
> > >
> > > Dear Reviewer en3y,
> > >
> > > Thank you for your continued engagement and for the helpful follow-up questions.
> > >
> > > ---
> > >
> > > ### **Response 1. Clarification on Table 5 comparisons (C-TAG vs. prior works)**
> > >
> > > Following the reviewer’s suggestion, we include in the revised Table 5 (p. 9) a **direct standalone comparison** between C-TAG and APG under the same SDXL setting:
> > >
> > > - Row “$+APG$” reports the standalone APG result (FID 19.523 with CLIPScore 26.71).
> > > - Row “$CFG_{C-TAG}$” reports the standalone our result (FID 19.798 with CLIPScore 26.30).
> > > - Row “$CFG_{C-TAG}$” with *windowing operation* (Sec. 6 (Limitations), p. 10) also reports the standalone our result (FID 19.288 with CLIPScore 26.23).
> > >
> > > We emphasize that TAG is fundamentally a sampler-level module rather than a modification of the CFG update. The core idea of TAG is to apply tangential re-weighting solely to the *solver update*, whereas other geometric guidance methods built on CFG (such as APG, TCFG, or CFG++) modify *how the conditional and unconditional terms are combined.* Consequently, TAG applies beyond these improved variants of CFG, including the regimes where CFG-based methods are not directly applicable, as well as the regimes where other guidance approaches such as PAG or SEG are applied
> > >
> > > To **keep the comparison protocol unambiguous**, we revised the paragraph following Table 5 (around lines 478–485 in the revised PDF) to explicitly state that the primary head-to-head evidence is the **standalone** comparison already shown in Table 5 (C-TAG vs. APG/TCFG), reported under identical prompts/seeds and the same SDXL setup.
> > >
> > > > Please note that with the introduction of the windowing strategy (detailed in Sec. 6), we re-conducted the full set of experiments to ensure a strictly fair comparison (using identical seeds and prompts across all models).

---

> ### Author Response · Authors · 2025-11-28
>
> ### **Response 2. Concern regarding “unusually poor baselines”**
>
> We agree that strong baselines and appropriate metrics are essential for interpreting improvements. For this reason, the revision places particular emphasis on **well-established conditional, high-fidelity regimes**, where both the baselines and evaluation protocol are standard:
>
> **For Table 4 (CLIPScore of CFG-based setting)**
>
> - **Contextualization of Table 4 Results**: Table 4 reports FID and CLIPScore of SD v1.5 with low CFG-scale (we used 2.5). This conservative setting was intentionally chosen to **isolate and emphasize** the fundamental effects of the TAG sampling mechanism, independent of strong classifier-free guidance signals.
> - **Expanded Evaluation in Practical Regimes**: **Appendix C (Practical High-Fidelity Regime, pp. 19-20)** expands our evaluation using **SD v2.1** with a **high CFG scale (= 7.5)**. We employ a broader metric suite—including **FID, CLIPScore, FD-DINOv2, and ImageReward**—to better capture practical trade-offs between low-level fidelity, semantic faithfulness, and geometric coherence.
> This section provides a deep analysis of TAG’s behavior in high-CFG regimes, emphasizing its hallucination-resistant capabilities (i.e., improved perceptual quality). Additionally, on the footnote of Table 4, **we explicitly clarify that Table 4 focuses on TAG's fundamental efficacy**, while **referring readers to Appendix C for its practical impact**.
>
> **For Table 3 (TAG with Higher-order solver)**
>
> - **Clarification**: In our experiments, we already used **DPM-Solver++ 2M** by default (noted as DPM++. We make it more clearly in the revision)
> - **Expanded Evaluation in DPM-Solver++ 2S**: Table 3 (p. 6) includes DPM++ variants (both 2S and 2M), and we observe that TAG improves FID/IS on these solvers as well, suggesting that the method is **not tied to a single first-order discretization**.
>
> **For Table 1, 2 (Unconditional Generation with StableDiffusion)**
>
> - **Reason of High-FID**: The higher FIDs (e.g., $\approx 100$) arise in the **unconditional** SD-family ImageNet evaluation, where performance is impacted not only by the **domain mismatch** between the SD training distribution and ImageNet reference statistics, but also by the absence of CFG in unconditional sampling, which can lead to **pronounced hallucination/artifact failures** in SD-family backbones.
> > This is a **known phenomenon in recent literature**; for instance, Table 1 of [A1] reports a **baseline SDXL FID of 129.5** for unconditional SDXL on ImageNet, confirming that high FIDs are expected in this specific setting.
> - **Expanded Evaluation in EDM2**: We report the standard low-FID regime (baseline FID 5.185) and evaluate TAG there as well (Table 7, p. 9), thereby contextualizing the unconditional SD results with a domain-aligned, high-fidelity backbone.
>
> ### **References**
>
> > [A1] Hong, Susung. "Smoothed energy guidance: Guiding diffusion models with reduced energy curvature of attention." Advances in Neural Information Processing Systems 37 (2024): 66743-66772. https://arxiv.org/html/2408.00760v2
>
> ---
>
> We hope that these clarifications on the standalone comparisons, together with the additional experiment results, help address your concerns and provide a clearer basis for reassessing our contribution.
>
> Sincerely,
> The Authors

---

### Official Review · Reviewer_d7Nb · 2025-10-31

**Soundness:** 3
**Presentation:** 3
**Contribution:** 4
**Rating:** 6
**Confidence:** 4

**Summary:**

This paper introduces Tangential Amplifying Guidance (TAG), a new inference-time guidance method for diffusion models based on the geometric decomposition of score updates.
TAG decomposes the diffusion update into radial and tangential components and amplifies only the tangential direction — which the authors argue carries the data-relevant semantic structure — while leaving the radial (noise) direction untouched.
This yields improved sampling fidelity, reduced hallucinations, and better semantic consistency without any architectural modification or retraining.

The paper provides:

• a theoretical justification (via Tweedie’s identity and Taylor expansion) that tangential amplification monotonically increases the local log-likelihood gain;

• an algorithmic formulation that is plug-and-play with standard samplers (DDIM, DPM++);

• and comprehensive experiments on multiple backbones (SD v1.5, SD v2.1, SDXL, SD3), both unconditional and conditional (CFG, PAG, SEG).

Overall, TAG achieves consistent improvements in FID, IS, and CLIPScore across settings, often outperforming baselines at fewer sampling steps.

**Strengths:**

Strong conceptual novelty.
The idea of separating the score update into radial/tangential components and amplifying only the tangential direction provides a geometric interpretation of guidance in diffusion models.

1. Solid theoretical analysis. The use of Tweedie’s identity and local log-likelihood expansion leads to a provable monotonic improvement theorem (Theorem 4.1).
The paper maintains a balance between geometric intuition and formal justification.

2. Empirical breadth and consistency. The authors evaluate across four diffusion families (DDIM, SD v1.5/v2.1, SDXL, SD3) and multiple guidance types (CFG, PAG, SEG).
Results are consistent and substantial, with up to ~20% FID reduction without additional NFE or retraining.

3. Plug-and-play practicality. TAG is architecture-agnostic, requires no finetuning, and adds negligible computational cost — a strong practical advantage.This makes it a likely method to be quickly adopted in practice.

4. Comprehensive ablations. The η ablation and the “avoidance of normal amplification” section clearly demonstrate the tradeoff between fidelity and over-smoothing, providing useful implementation guidance.

5. Readable and well-organized. The paper is clearly written, with consistent notation and good figures illustrating the geometric intuition.

**Weaknesses:**

1. Fixed amplification factor (η).The method uses a global η for all timesteps. While the authors note in §6 that adaptive ηₖ could further improve performance, the current approach might not optimally handle varying signal-to-noise ratios across timesteps. A simple experiment on dynamic ηₖ would strengthen the work.

2. Limited discussion of generalization beyond images. TAG is conceptually general, but all experiments are on 2D image diffusion.
A brief experiment (even qualitative) on another modality (e.g., audio, latent diffusion for text, or 3D) could make the impact broader.

**Questions:**

None

---

> ### Author Response · Authors · 2025-11-21
>
> **Dear reviewer d7Nb,**
>
> We sincerely appreciate your thoughtful review and constructive feedback. Below we respond to your comments and point to the corresponding updates in the revised manuscript.
>
> ---
>
> ### **Comment 1**
>
> > **Fixed amplification factor $\eta$... a simple experiment on dynamic $\eta_k$ would strengthen the work.**
>
> **Response 1**
>
> We agree that using a single global amplification factor $\eta$ across all timesteps can be sub-optimal, since the reliability of the tangential/radial decomposition and the SNR vary over the diffusion trajectory.
> Motivated by your suggestion, we added a **timestep-dependent variant of TAG** in the revised manuscript (Section 6, Page 10).
>
> Concretely, we consider a **simple windowed schedule**:
>
> * TAG is applied only within a diffusion-timestep window ($[\eta_{\text{sta}}, \eta_{\text{end}}]$);
> * outside this window, we set $\eta_{k} = 1$ (i.e., no tangential amplification).
>
> This yields a **piecewise-constant dynamic $\eta_{k}$** that turns TAG on where the Gaussian-annulus / spherical-shell approximation is most reliable (moderate–high noise), and off in the low-noise regime where tangential estimates can be less aligned (we added new discussion in Sec. 6). Importantly, this schedule introduces **no additional model evaluations or learnable parameters**, and serves as a minimal yet meaningful dynamic $\eta_{k}$ baseline.
>
> The results are summarized in **Table 8 (Page 10)**. On ImageNet with SD v1.5:
>
> * Restricting TAG to the **low-noise window** ([400,0]) gives the lowest quality.
> * Applying TAG over the **entire trajectory** ([1000,0]) improves quality but is not optimal.
> * Applying TAG only in a **mid/high-noise window** ([1000,400]) yields the best performance, improving FID by $\approx 6$% at the same NFEs.
>
> These results directly support your point: a **timestep-dependent $\eta_{k}$** that respects the changing noise regime can outperform a globally fixed $\eta$. While this is a simple on/off schedule, it already captures non-uniform SNR effects; we view more adaptive schedules (e.g., learned or score-dependent $\eta_{k}$) as a promising extension beyond this initial study.
>
> ---
>
> ### **Comment 2**
>
> > **Limited discussion of generalization beyond images...**
>
> **Response 2**
>
> We agree, and we added a **qualitative beyond-2D experiment** to demonstrate TAG’s generality. Specifically, we now include an **Image-to-3D Shape Generation** experiment using **SV3D** [A1] (revised manuscript, Figure 10, Page 10). TAG is inserted during SV3D’s diffusion-based novel-view synthesis, and it yields **improvements in preserving semantic details** relative to the baseline, without any architectural change.
>
> While this is an initial qualitative study, it provides **early evidence** that TAG can transfer to higher-dimensional generative settings. We will continue exploring other modalities (e.g., audio/video) in future work.
>
> ### **References**
>
> > [A1] Voleti, Vikram, et al. "Sv3d: Novel multi-view synthesis and 3d generation from a single image using latent video diffusion." ECCV 2024.
>
> ---
>
> We again sincerely thank you for these insightful suggestions, which directly motivated the new timestep-dependent experiments and the 3D generalization study, strengthening the revised paper.

---

> > ### Author Response · Authors · 2025-11-28
> > **Kind reminder (Paper #14964)**
> >
> > Dear Reviewer d7Nb,
> >
> > Thank you again for your time and effort in reviewing our manuscript. We have posted our response addressing your concerns and suggestions.
> >
> > If you have any additional questions or require further clarification, we are happy to discuss them. We eagerly await your valuable feedback.
> >
> > Best regards,
> >
> > Authors of Submission #14964

---

### Official Review · Reviewer_rNL6 · 2025-11-01

**Soundness:** 3
**Presentation:** 3
**Contribution:** 3
**Rating:** 4
**Confidence:** 4

**Summary:**

The work proposes to separate the sampling denoising process into two direction at each timestep. One direction is normal representing how noisy the data is and one direction is tangential representing the main content of the image. The work proposes a guidance to enhance tangential direction so that improves the fidelity without introducing external information. The results show some significant improvement.

**Strengths:**

1. The paper is well written and easy to understand
2. The proposed method is intuitive and well motivated theoretically

**Weaknesses:**

The main concerns lie in the baselines of the paper.

1. Most of the baselines in the paper are not very consistent with what we have in the literature. Most of the FID score is around 100 which is too high compared to all the literatures we have known which is around 1 to 10 in ImageNet. The CLIP scores from SDv2.1 is also too low 22.x which is significant lower than the number reported in other papers around 30.x
2. The baselines in the paper are from DDIM/DDPM which are too old. While these baselines are useful, we would want to expect to see newer baselines such as EDM-2, Flux, Flow matching models which are not reported in the paper. In table 5, we have flow matching number but the FID is 150 for ImageNet, this number is too high to tell if the generated data is useful.

**Questions:**

Please see the weaknesses

Please consider making the reported numbers more consistent with the literature, so that we can see what it really can improve. Otherwise, improvement on poor baselines make no sense in practice. I will increase the score after fixing the concerns.

---

> ### Author Response · Authors · 2025-11-21
>
> **Dear reviewer rNL6,**
>
> Thank you for your careful reading and for clearly pointing out the baseline issue. We agree that the strength and modernity of baselines are crucial for interpreting TAG’s practical value. Below we respond to your two connected concerns.
>
> ---
>
> ### **Comment 1 & 2**
>
> > **C1. Most baselines are not consistent with the literature (FID $\approx$ 100 for unconditional, CLIP $\approx$ 22 vs $\approx$ 30 for conditional).**
> > **C2. Baselines are too old (DDIM/DDPM); please include newer backbones such as EDM-2 / Flow-matching. The FID≈150 flow-matching baseline is too high to judge usefulness.**
>
> **Response 1 & 2**
>
> We agree that improvements on unusually weak baselines would be uninformative. To address this, we made two revisions:
>
> **(1) Clarification on the original scale of FID/CLIP.**
> - The high FIDs (~100) in our original submission primarily came from **unconditional ImageNet generation using SD-family backbones**, which are not ImageNet-specialized diffusion models and therefore yield substantially weaker baselines than works reporting FID 1–10 on ImageNet-tuned backbones.
>
> - The relatively lower CLIPScore in our conditional results arose because we also evaluated **low-CFG and condition-only regimes on MS-COCO**, which are known to be weaker than the high-CFG settings typically used in recent text-to-image literature.
>
> To address these issues, we added stronger baselines and evaluation protocols aligned with recent literature.
>
> **(2) Added modern, literature-faithful baselines.**
> Following your suggestion, we added stronger modern backbones and re-evaluated the flow-matching baseline with a realistic protocol:
>
> * **EDM2 [A1] (modern ImageNet backbone).**
>   We added experiments on **EDM2** using its **official ImageNet-512 generation/evaluation pipeline**. The EDM2 baseline is already literature-aligned, and TAG remains beneficial on top of this strong backbone, consistently improving semantic quality (FD-DINOv2) while largely preserving FID. (Table 7, Page 9)
>
> * **SD3 [A2] (flow-matching).**
>   In the revision, we re-evaluate unconditional SD3 on ImageNet with a **30K-sample protocol and matched 50 NFEs**. TAG still improves this baseline at the same compute. (Table 6, Page 9)
>
> * **SDXL [A3] (high-CFG regime).**
>   We expanded SDXL experiments under high-CFG settings (**$\omega = 5.0$**, 50 steps). TAG continues to provide consistent improvements on top of these stronger baselines. In particular, as illustrated in Figures 13-16 (Pages 23–26), the qualitative results show TAG working as a hallucination-reduction mechanism, confirming that TAG remains applicable in modern, practical setups.
>
> We keep DDIM/DDPM results as widely used reference samplers, but the revised manuscript now also verifies TAG on **modern backbones (EDM2, SD3 flow-matching, SDXL)** with baselines aligned to current literature scales. We hope these experiments address your concern.
>
> ### **References**
> > [A1] Karras, Tero, et al. "Analyzing and improving the training dynamics of diffusion models." Proceedings of the IEEE/CVF Conference on Computer Vision and Pattern Recognition. 2024.
> > [A2] Esser, Patrick, et al. "Scaling rectified flow transformers for high-resolution image synthesis." Forty-first international conference on machine learning. 2024.
> > [A3] Podell, Dustin, et al. "Sdxl: Improving latent diffusion models for high-resolution image synthesis." arXiv preprint arXiv:2307.01952 (2023).
>
> ---
>
> We sincerely thank you again for pushing us to strengthen the baselines. Your comments directly motivated the added EDM2, SDXL, and SD3 flow-matching experiments and a clearer calibration of our empirical claims.

---

> > ### Author Response · Authors · 2025-11-28
> >
> > **Dear Reviewer rNL6,**
> >
> > We have further revised the manuscript to alleviate your concerns regarding Table 4.
> >
> > ---
> >
> > **For Table 4 (CLIPScore of CFG-based setting)**
> >
> > - **Expanded Evaluation in Practical Regimes**: **Appendix C (Practical High-Fidelity Regime, pp. 19-20)** expands our evaluation using **SD v2.1** with a **high CFG scale (= 7.5)**. We employ a broader metric suite—including **FID, CLIPScore, FD-DINOv2, and ImageReward**—to better capture practical trade-offs between low-level fidelity, semantic faithfulness, and geometric coherence.
> > This section provides a deep analysis of TAG’s behavior in high-CFG regimes, emphasizing its hallucination-resistant capabilities (i.e., improved perceptual quality). Additionally, on the footnote of Table 4, **we explicitly clarify that Table 4 focuses on TAG's fundamental efficacy**, while **referring readers to Appendix C for its practical impact**.
> >
> > ---
> >
> > We are grateful again for your feedback that motivated a stronger baseline evaluation.

---

### Author Response · Authors · 2025-12-01
**Author Final Remarks**

**Dear Area Chair and Senior Area Chair,**

Thank you for coordinating the review process. As the discussion closes, we summarize the paper’s core contribution and how the revision addresses the decision-critical concerns.

---

## Summary of Work

We **provide theoretical and empirical evidence** that semantic refinement is **more aligned with** the tangential component along iso-noise surfaces, while the radial component primarily reflects noise-level progression.

We therefore propose TAG, which selectively amplifies the tangential component to reduce hallucinations in a plug-and-play manner, with no additional NFEs.

---

## I. Consensus on contribution & technical soundness

**Reviewers broadly converged** on the technical soundness and contribution of TAG.

- **Strong Support**: **RZdQ (8)** and **d7Nb (6)** emphasized the **"solid theoretical analysis,"** citing **Theorem 4.1** (monotonic improvement of the first-order Taylor gain—our local likelihood proxy) and its **plug-and-play** nature.
- **Convergence after clarifications**: **rNL6** praised the method as **"intuitive and well motivated theoretically."** Most notably, en3y initially raised a novelty concern by viewing TAG as an APG **[2]**-like projection variant. After we clarified that TAG is a solver-level, condition-agnostic geometric modification rather than a CFG-space projection (**Appendix B**), **en3y acknowledged that the contribution ”now seem to be technically valid”**. This acknowledgement underscores TAG’s novelty and broad applicability **beyond CFG-dependent projection** methods.

---

## II. Empirical evidence and strengthened evaluation

**RZdQ (8)** and **d7Nb (6)** supported TAG’s practical value; **plug-and-play gains without additional NFEs.**

**Key concerns raised (rNL6, en3y):** reviewers requested (i) evaluation in **literature-faithful high-fidelity regimes**, and (ii) **standalone, fully controlled head-to-head comparisons** against the closest related baselines.

**Revision on unconditional generation (i)**

- **Clarification:** Prior work reports similarly **high FIDs for unconditional SDXL** on ImageNet (e.g., [1]).
- **Modern backbone:** To align with literature-faithful ImageNet protocols, we additionally evaluate TAG on the official EDM2 ImageNet-512 pipeline (Table 7). EDM2 baseline FID is 5.185 (i.e., in the 1–10 range), and **TAG improves it to 5.164**.

**Revision on conditional generation (i), (ii)**

- **High-fidelity regime I:** To address rNL6 and en3y’s remaining concern regarding **literature-faithful high-fidelity settings**, **Appendix C** reports SD v2.1 evaluation at **high CFG (7.5)** with a broader metric suite and **paired** analysis. Notably, TAG yields **up to +19.6% relative ImageReward** even in this high-CFG regime.
- **High-fidelity regime II**: Following **en3y**’s request, **Table 5** reports standalone comparisons under matched settings against the closest **projection-based** baselines (**SDXL-based APG and TCFG**); additional qualitative results are in **Appendix F.1**.

---

## III. Applicability and limitations

Reviewers suggested adding (i, d7Nb) further ablations to assess robustness and limitations (including $\eta$ sensitivity), (ii, RZdQ) practical overhead measurements (latency/VRAM), and (iii, d7Nb) evidence of broader applicability beyond standard 2D image generation.

- **Mitigating $\eta$ sensitivity:** We introduced a **windowed $\eta$** schedule (Table 8), further improving performance.
- **Compute, Latency and Memory overhead:** **Appendix D (Table 12)** reports compute/latency and memory, showing TAG preserves baseline VRAM and adds only modest latency overhead.
- **Generality beyond 2D images:** We added **Image-to-3D** qualitative results (Fig. 10) demonstrating that TAG’s geometric modification has the potential to be transferred beyond standard 2D image generation.

---

## IV. Conclusion

- Reviewers broadly converged on the work’s technical soundness, and several praised its theoretical grounding (e.g., **RZdQ, d7Nb, rNL6**).
- We strengthened the empirical evidence by clarifying the high-FID scale in unconditional SD-family ImageNet settings, extending evaluation to literature-faithful high-fidelity regimes, and adding standalone, controlled head-to-head comparisons against projection-based baselines.
- We further mitigated $\eta$-sensitivity via a simple windowed schedule and provided preliminary evidence of transfer beyond 2D images.

**Reference**

> [1]: https://arxiv.org/abs/2408.00760, NeurIPS24
[2]: https://arxiv.org/abs/2410.02416, ICLR25

---

Best regards,

The Authors of #14964.

---

### Meta-Review · Area_Chair_xJ1b · 2025-12-22

**Summary:**

The major concerns raised by the reviewers are the use of outdated baselines and limited comparison to more recent works, the inconsistency of the performances of baselines in literature, and the novelty of the proposed framework compared to closely related prior work. Other concerns include hyper-parameter selection, generalizability beyond 2D image generation, notation confusion, and lack of qualitative visual examples.

**Reviewer Concerns:**

Some concerns were addressed during the rebuttal, such as the unusually low baseline performance, generalization beyond 2D images, hyper-parameter selection, notation confusion, and lack of visual examples. The major concerns about missing baselines were not fully addressed. Even though additional experiments were added during the rebuttal, some reviewers still have concerns about the strength and consistency of the baselines and the fairness of comparison setup. A more thorough experimental study, which takes longer time, would be necessary to fully resolve the issues.

**Reviewer Scores:**

Reviewer rNL6 may increase the score. Reviewer d7Nb and RZdQ may keep the positive score. However, Reviewer en3y is likely to keep the negative score.

---

### Decision · Program_Chairs · 2026-01-26

Reject